# Simulations of the impact of CCN and INP perturbations on the microphysics and radar reflectivity factor of stratiform mixed-phase clouds

Junghwa Lee [1], Patric Seifert [1], Tempei Hashino [2], Maximilian Maahn [3], Fabian Senf [1], and Oswald Knoth [1]

[1]Leibniz Institute for Tropospheric Research (TROPOS), Leipzig, Germany
[2]Kochi University of Technology, Kochi, Japan
[3]Leipzig University, Leipzig, Germany

**Correspondence:** Junghwa Lee (lee@tropos.de)

**Abstract.** In this research, we delve into the influence of cloud condensation nuclei (CCN) and ice-nucleating particles (INP) concentrations on the morphology and abundance of ice particles in mixed-phase clouds, emphasizing the consequential impact of ice particle shape, number, and size on cloud dynamics and microphysics. Leveraging the synergy of the Advanced Microphysics Prediction System (AMPS) and the Kinematic Driver (KiD) model, we conducted simulations to capture cloud microphysics across diverse CCN and INP concentrations. The Passive and Active Microwave radiative TRAnsfer (PAMTRA) radar forward simulator further augmented our study, offering insights into how the concentrations of CCN and INP affect radar reflectivities.

Our experimental framework encompassed CCN concentrations ranging from 10 to 5000 $cm^{-3}$ and INP concentrations from 0.001 to 10 $L^{-1}$. Central to our findings is the observation that higher INP concentrations yield smaller ice particles, while an increase in CCN concentrations leads to a subtle growth in their dimensions. Consistent with existing literature, our results spotlight oblate-like crystals as dominant between temperatures of $-20$ to $-16$ °C. Notably, high INP scenarios unveiled a significant prevalence of irregular polycrystals. The Aspect Ratio (AR) of ice particles exhibited a decline with the rise in both CCN and INP concentrations, highlighting the nuanced interrelation between CCN levels and ice particle shape, especially its ramifications on the riming mechanism.

The forward-simulated radar reflectivities, spanning from $-11.83$ dBZ (low-INP, 0.001 $L^{-1}$) to 4.65 dBZ (high-INP, 10 $L^{-1}$), elucidate the complex dynamics between CCN and INP in determining mixed-phase cloud characteristics. Comparable differences in radar reflectivity were also reported from observational studies of stratiform mixed-phase clouds in contrasting aerosol environments. Our meticulous analysis of KiD-AMPS simulation outputs, coupled with insights into aerosol-driven microphysical changes, thus underscores the significance of this study in refining our ability to understand and interpret observations and climate projections.

# 1 Introduction

Clouds are still one of the most uncertain components of the global atmosphere system (Bony et al., 2015). Their formation and evolution occur on various spatio-temporal scales, which makes it virtually impossible to tackle them with single, unified observational or simulation approaches (Kahn et al., 2023). Single sub-processes shall be studied individually and will only in a later stage be the basis for an improved comprehensive understanding. Important components of a cloud's life cycle are, e.g., the cloud formation and the subsequent transitions from the liquid to the ice phase. The presence of the ice phase is an essential prerequisite for the production of adequate amounts of precipitation in most regions on Earth (Mülmenstädt et al., 2015). Nevertheless, for the initial formation of cloud droplets, as well as for the formation of ice crystals down to temperatures of $-38°C$, aerosol particles are required for the phase transition by providing either a reservoir of cloud condensation nuclei (CCN) or ice nucleating particles (INP), respectively (Morrison et al., 2012; Hoose and Möhler, 2012). The interplay of the abundance of CCN and INP and the cloud evolution is an important pathway of aerosol-cloud interaction. Especially perturbations in the concentration and type of CCN or INPs can potentially influence the formation and evolution of ice particles in mixed-phase clouds.

There are strong indications given by both, observations and modeling approaches, that INP and CCN perturbations do have a considerable impact on the mixed-phase cloud formation and evolution. Seifert et al. (2010) revealed increased fractions of ice-containing clouds in dust-laden cloud environments over Central Europe. On a global scale, these findings were confirmed, e.g., by Zhang et al. (2018), using observations from the space-borne A-Train satellite constellation. Hemispheric contrasts in mixed-phase clouds and their relationship to cloud turbulence and aerosol load were investigated in detail by Radenz et al. (2021), who also concluded that a measurable impact of aerosol on mixed-phase cloud formation exists. Seifert et al. (2012) revealed considerable impacts of a dust event on the simulation of clouds and precipitation patterns over Germany. Similar effects were identified in a European-scale approach by Barthlott and Hoose (2018). Fan et al. (2014) performed spectral bin simulations to investigate the influence of CCN and INP on precipitation in two distinct mixed-phase orographic cloud scenarios characterized by different cloud temperatures. The study revealed varying degrees of significance regarding the impacts of CCN and INP on precipitation, with the INPs exhibiting a more pronounced effect in both cases. Furthermore, Fan et al. (2017) conducted a sensitivity analysis where they systematically varied the concentrations of CCN and INP proxies across a wide range, spanning from extremely low to extremely high concentrations, employing spectral bin modeling specifically tailored for orographic mixed-phase clouds.

Also on a global scale, aerosol variations were found to be key for understanding the variability of mixed-phase clouds (Atkinson et al., 2013). Recently, even the first closure studies bridging remote-sensing observations of CCN and INP with those of cloud droplet concentration and ice crystal number concentration were brought underway (Ansmann et al., 2019; Engelmann et al., 2021). However, simulations and observational approaches were to date rarely combined, which hinders one from drawing specific conclusions on aerosol effects on mixed-phase clouds. One key approach is to connect cloud-resolving, aerosol-sensitive, ideally spectral-bin, models with forward operators in order to transfer simulation output into observation space. By doing so, simulations for selected scenarios can be evaluated against real-world observations.

Given the complexity of spectral-bin modeling frameworks, it is essential to incorporate the most relevant processes on the one hand, but to constrain the environmental conditions to a maximum but still realistic state, on the other hand. Besides number concentration, thus also particle habit should be incorporated into respective aerosol-cloud-interaction studies which aim at a closure against observations. Ice particle shape plays a crucial role in determining the microphysical and radiative properties of mixed-phase and ice clouds (Mishchenko et al., 1996; McFarquhar and Heymsfield, 1997). The diverse shapes of ice particles influence their growth, aggregation, and riming processes, which in turn affect cloud lifetime, precipitation formation, and radiative energy transfer within the atmosphere (Magono and Lee, 1966; Heymsfield and Westbrook, 2010; Um and McFarquhar, 2011). The complexity and diversity of ice particle shapes present challenges for both cloud microphysics modeling and remote sensing of cloud properties. A comprehensive understanding of ice particle shape is essential for improving the accuracy of cloud microphysics models, remote sensing retrievals, and ultimately, climate predictions (Liou and Ou, 2004; Tao et al., 2012; Chen and Liu, 2016; Vázquez-Martín et al., 2021). Despite its importance, the representation of ice particle shape in cloud microphysics models remains a significant challenge. Many models adopt simplified assumptions regarding ice particle shape, such as assuming all particles are spherical or using a limited set of predefined shapes (Mitchell, 1996; Morrison et al., 2005; Cotton et al., 2013). These simplifications can introduce uncertainties and biases in the simulated cloud properties and their interactions with radiation (Cotton et al., 2013; Eriksson et al., 2015). Furthermore, the complex nature of ice particle shape and its dependence on factors such as temperature, supersaturation, and aerosol loading adds to the difficulty in accurately representing this aspect of cloud microphysics (Bailey and Hallett, 2009; Kanji et al., 2017).

Radar remote sensing is a valuable tool for observing ice particles in clouds, providing insights into their size, shape, and spatial distribution (Hogan et al. 2000; Westbrook and Illingworth, 2011). However, interpreting radar observations of ice particles requires a thorough understanding of the relationship between ice particle shape and the radar variables, such as reflectivity and Doppler velocity (Hogan et al., 2012; Kneifel et al., 2015). Radar forward simulators, which generate synthetic radar observations based on cloud model outputs, can help bridge this gap by allowing researchers to systematically investigate the sensitivity of radar variables to different ice particle shapes and model assumptions (Matsui et al., 2019).

In this study, we utilize the Advanced Microphysics Prediction System (AMPS) coupled with the Kinematic Driver (KiD) to conduct idealized simulations of mixed-phase cloud microphysics (Hashino and Tripoli 2007; Hashino and Tripoli 2008; Hashino and Tripoli 2011a; Hashino and Tripoli 2011b), incorporating a comprehensive representation of ice particle shapes and the effects of CCN and INP perturbations. AMPS is a state-of-the-art cloud microphysics model that has been specifically designed to capture the complex interactions between aerosols, cloud droplets, and ice particles with a habit prediction system (Hashino and Tripoli 2007; Hashino and Tripoli 2008; Hashino and Tripoli 2011a). The AMPS model coupled with large-eddy simulations (LES) successfully reproduces features of mixed-phase clouds and has been compared to observations (Hashino et al., 2020; Ong et al., 2022). To investigate the impact of ice particle shape on radar retrievals, we employ the Passive and Active Microwave radiative TRAnsfer (PAMTRA) radar forward simulator. PAMTRA is a versatile tool that can simulate passive and active microwave observations of the atmosphere, accounting for the scattering properties of various ice particle shapes (Mech et al., 2020). By combining the capabilities of AMPS and PAMTRA, this study aims to provide a comprehensive understanding of the role of ice particle number size distribution and shape in mixed-phase cloud microphysics and remote sensing retrievals

under varying CCN and INP conditions. Furthermore, we seek to evaluate the impact of ice particle shape assumptions and CCN/INP perturbations on the accuracy and reliability of cloud property retrievals from radar observations.

This paper is organized as follows. Sect. 2 briefly describes the Kinematic Driver (KiD) (Shipway and Hill, 2012) as the dynamical model and AMPS as the microphysics model. Sect. 3 gives information on the initial thermodynamic condition and experimental design for simulations. Sect. 4 shows the numerical simulation results for steady-state mixed-phase cloud cases under varying CCN and INP scenarios. Finally, Sect. 5 concludes the paper with a summary of the results.

## 2 Model description and simulation setup

### 2.1 KiD (dynamic model)

The KiD model provides a framework for examining cloud microphysics, enabling us to assess and compare different parameterizations, which leads to a better understanding of cloud particle interactions and the influence of aerosols on cloud development. The model's versatility allows its application in the study of various cloud types, including stratiform mixed-phase and convective clouds. It has significantly contributed to the enhancement of cloud microphysical parameterizations in larger-scale models (Klein et al., 2009; Shipway and Hill, 2012).

The model accommodates a variety of microphysics schemes, from the simpler 1-moment bulk models (Thompson et al., 2004) to more complex 2-moment schemes (Thompson et al., 2008; Morrison et al., 2009; Shipway and Hill, 2012; Hill et al., 2015; Vié et al., 2016; Miltenberger et al., 2018). It is also compatible with detailed spectral-bin microphysics schemes, including the Tel-Aviv University bin microphysics and AMPS (Tzivion et al., 1987, 1989; Hashino and Tripoli, 2007, 2008, 2011a, b; Lebo and Seinfeld, 2011; Onishi and Takahashi, 2012), as well as with Lagrangian Cloud Models (LCMs) (Andrejczuk et al., 2010; Arabas et al., 2015; Hoffmann et al., 2015; Dziekan et al., 2019). Further details on these aspects are provided in Shipway and Hill (2012) and Hill et al. (2023).

We emphasize the KiD framework's effectiveness in efficiently evaluating the performance of different microphysics schemes. Additionally, the KiD model functions as a valuable benchmarking tool, enabling researchers to evaluate and enhance cloud microphysics parameterizations. By comparing the results of various parameterizations within the KiD framework, inconsistencies and areas for improvement can be identified and investigated.

### 2.2 AMPS (microphysics model)

In this study, we employed the Kinematic Driver model (KiD) in conjunction with the Advanced Microphysics Prediction System (AMPS) to simulate mixed-phase clouds. The AMPS model has been coupled with other dynamic models such as the University of Wisconsin-Nonhydrostatic Modeling system (Hashino and Tripoli, 2007, 2008; Hashino et al., 2020) and the Scalable Computing for Advanced Library and Environment (SCALE) large-eddy simulation model (Ong et al., 2022), demonstrating its capability to accurately predict mixed-phase clouds and exhibiting favorable comparisons with observational data.

**Table 1.** List of the 16 particle property variables (PPVs) within liquid and ice spectra across each bin, as utilized in the KiD-AMPS model. Here, $\rho_m$ denotes the moist air density, while $\rho_{\text{lat}}$ and $\rho_{\text{iat}}$ specify the total aerosol density within the liquid phase and ice phase, respectively. Additionally, $\rho_{\text{las}}$ and $\rho_{\text{ias}}$ correspond to the soluble aerosol density in the liquid and ice phases.

| Spectrum | PPV | Description |
|---|---|---|
| Liquid | $\rho_{\text{lat}}/\rho_m$ | Mixing ratio of total aerosol mass |
| | $\rho_{\text{las}}/\rho_m$ | Mixing ratio of soluble aerosol mass |
| Ice | $\rho_{\text{cry}}/\rho_m$ | Mixing ratio of crystal mass |
| | $\rho_{\text{rim}}/\rho_m$ | Mixing ratio of riming mass |
| | $\rho_{\text{agg}}/\rho_m$ | Mixing ratio of aggregate mass |
| | $\rho_{\text{frz}}/\rho_m$ | Mixing ratio of frozen mass |
| | $\rho_{\text{mlt}}/\rho_m$ | Mixing ratio of meltwater mass |
| | $\rho_{\text{iat}}/\rho_m$ | Mixing ratio of total aerosol mass |
| | $\rho_{\text{ias}}/\rho_m$ | Mixing ratio of soluble aerosol mass |
| | $n_{\text{exice}}$ | Extra crystalline structure number |
| | $V_{\text{cs}}$ | Circumscribing volume |
| | $l_a^3$ | Cube of the a-axis length |
| | $l_c^3$ | Cube of the c-axis length |
| | $l_d^3$ | Cube of the d-axis length (Cube of the dendritic arm) |
| | $a_g$ | Center of gravity along the a-axis |
| | $c_g$ | Center of gravity along the c-axis |

According to Hashino and Tripoli (Hashino and Tripoli, 2007, 2008, 2011a, b), the AMPS microphysical model employs the Spectral Ice Habit Prediction System (SHIPS), which incorporates particle property variables (PPVs) to characterize the physical structure of ice particles, as detailed in Table 1. The SHIPS continuously updates the PPVs for each mass bin in response to evolving ambient conditions, ensuring accurate particle property diagnoses based on the PPVs.

The identification of ice particle type and habit relies on various components, including mass content, length, and concentration. In our methodology, the SHIPS defines the ice particle model as a conceptual shape to represent ice particles leading to their genesis, encompassing "pristine crystals, aggregates, rimed aggregates, graupel, and rimed crystals" as explicated in Figure 2 by Hashino and Tripoli (2007). Pristine crystals, rimed crystals, aggregates, and rimed aggregates are modeled as cylinders, while graupel is represented as a spheroid. These shapes serve as the basis for determining the maximum dimension

(D) of each particle. However, it is important to note that this diagnosis is primarily intended for comparison with observations or other models using predicted mass bin information within the PPVs. Therefore, additional errors may arise when artificially categorizing these types. In this study, we analyzed the mass bin information without separate type divisions for a more comprehensive assessment.

     The habit of ice crystals, a critical aspect of our study, is determined by analyzing their unique crystallographic properties

which include forms such as plates, dendrites, columns, and three polycrystals, as illustrated in Figure 1. For each identified

particle habit, the SHIPS within the AMPS model assigns an ice particle model that represents the geometric shape enveloping the ice particle. This model encompasses detailed crystal habit information—such as the a-axis length ($l_a$), representing the radius; the c-axis length ($l_c$), representing the height; and the d-axis length ($l_d$), representing the dendritic arm—across the three crystal habits of plate, columnar, and dendrite for monocrystals. Additionally, the model employs a PPV, termed the

140 extra crystalline structure number ($n_{exice}$), which ranges from 0 to 1. A value of $n_{exice}$ greater than or equal to 0.5 signifies that the ice crystals in a particular bin are polycrystals. The SHIPS's methodical approach also integrates the coordinates of the center of gravity ($a_g$, $c_g$), measured along the a- and c-axes from the center of the monocrystals, as distinct PPVs. These measurements are pivotal in differentiating between planar and columnar polycrystals: a planar polycrystal is identified if the ratio $a_g/l_a$ exceeds the ratio $c_g/l_c$ by more than 0.5, whereas a columnar polycrystal is determined if $c_g/l_c$ exceeds $a_g/l_a$ by more

than 0.5. Ice crystals that do not fit within these criteria, such as scale-like side planes, are categorized as irregular polycrystals. The process and criteria for habit diagnosis are further detailed in Figure 2.

To explain more about ice particle habit, the ice crystal growth regime is determined based on the cumulative relative frequency of habits and a random number generator when the temperature falls below -20°C and the maximum dimension of the crystal is less than 20 $\mu$m. The growth regime is initially selected from polycrystalline, columnar, and planar hexagonal

regimes. If the growth regime is polycrystalline, a random number determines whether it is columnar or planar polycrystalline. Furthermore, if it is polycrystalline, another random number determines the growth regime for a hexagonal monocrystal. This implies that small ice crystals can grow differently from the habit diagnosed at the beginning of the time step. Conversely, for temperatures above -20°C, it is assumed that polycrystals do not form. Once the maximum dimension exceeds 20 $\mu$m, the ice crystal is presumed to follow the growth of the diagnosed habit at the beginning of the time step.

Expanding upon this, the concept of AR of an object is the ratio of its width to its height. In this study, the AR of polycrystalline ice particles is determined by the ratio of the semi-axis lengths of the ice particle model, i.e., $\alpha = l_{c,sm} / l_{a,sm}$, while the AR of monocrystalline particles is determined by the ratio of the axis lengths, $\alpha = l_c / l_a$.

Within the framework of the AMPS model, ice particles are characterized by their circumscribing sphere volume, denoted as $V_{cs}$. This volume is pivotal for comprehending the microphysical behavior of ice particles within mixed-phase clouds. As

Hashino and Tripoli (2011a) explain, the circumscribing sphere volume of an ice particle is instrumental in forecasting the mass-dimension (m-D) relationship. The AMPS model utilizes functions that interlink the predicted sphere volume with the diagnosed aspect ratios, semi-axis lengths, and the particle's maximal extent. Each mass bin's circumscribing sphere volume, a key PPV, is integral to these prognostications. The model assumes a consistent geometric form across microphysical processes, transferring the concentration-weighted circumscribing sphere volume between mass bins as per the collection process. This

transfer is essential to ascertain the circumscribing sphere volume of representative hydrometeors, thereby ensuring an accurate representation of the ice particles' physical properties. Moreover, this factor critically influences the determination of the effective diameter. The formulation for calculating $V_{cs}$ depends on the particle type: for most ice particles, it is determined by the equation $k_s(l_{a,sm})^3(1+\alpha^2)^{3/2}$ assuming a cylinder, where $k_s = 4\pi/3$, and $\alpha$ represents the aspect ratio (AR) of the ice particle defined as the ratio of the vertical semiaxis length ($l_{c,sm}$) to the horizontal semiaxis length ($l_{a,sm}$). However, for

graupel, it is computed using $k_s(l_{a,sm})^3 \max(1,\alpha^3)$ assuming an ellipsoid.

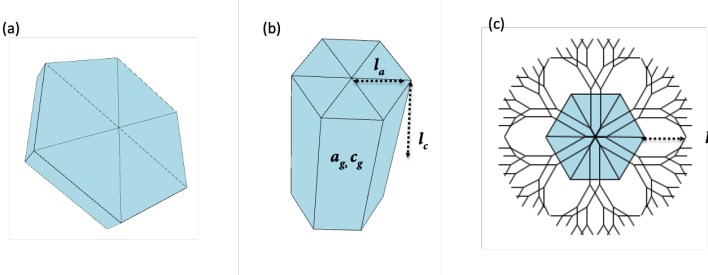

**Figure 1.** Diagnosis of the habit of the representative hydrometeor dimensions ($l_a$, $l_c$, and $l_d$) for monocrystals, including (a) hexagonal plate, (b) column, and (c) dendrite.

It is important to highlight that while categorizing solid hydrometeors into specific types and habits is not obligatory for conducting SHIPS microphysics simulations, it greatly improves the model verification process. This is especially valuable because observational data are frequently organized based on these conventional classifications, facilitating more robust model comparisons. Given this capability, we conclude that AMPS is well-suited to investigate the impact of varying CCN and INP conditions on particle shape.

### 2.3 PAMTRA (Radar forward Simulator)

The equivalent radar reflectivity factor denoted as $Z_e$, characterizes the collective scattering properties of a volume of scatterers, such as atmospheric precipitation particles, rather than just a single object. This factor is crucial in identifying and quantifying precipitation events by enabling the detection of radar signal reflections from these particles. The magnitude of $Z_e$ is influenced by several factors: the size and concentration of the precipitation particles, their composition, and the frequency of the radar signal.

To convert the model data into radar variables, we utilize PAMTRA, a powerful tool designed for simulating and retrieving microwave radiative properties in the atmosphere. Serving as a forward model, PAMTRA allows the interpretation of data from diverse passive and active microwave sensors, including radars and radiometers, used for observing and studying precipitation and other atmospheric phenomena. PAMTRA incorporates crucial factors such as cloud and precipitation scattering, and gas absorption, and facilitates comprehensive analysis of remote sensing data from satellite, airborne, and ground-based platforms. Numerous studies have leveraged PAMTRA to investigate precipitation processes, cloud microphysics, and remote sensing of atmospheric variables, establishing it as an invaluable resource for researchers and atmospheric scientists engaged in the field of radiative transfer and remote sensing (Maahn et al., 2019; Ori et al., 2020; Schnitt et al., 2020; von Lerber et al., 2022).

PAMTRA offers a full-bin interface with several benefits, which can directly convert spectral-bin model data or in situ measurement for each size bin to radar variables. This conversion facilitates the transfer of crucial information, such as mass, density, number concentration, terminal velocity, cross-sectional area, AR, and particle size distribution, without the need for further assumptions. In contrast, a bulk microphysics model provides hydrometeor mixing ratio in one-moment microphysics models (e.g., Baldauf et al., 2011) or mixing ratio and number concentration in two-moment microphysics models (e.g., Seifert

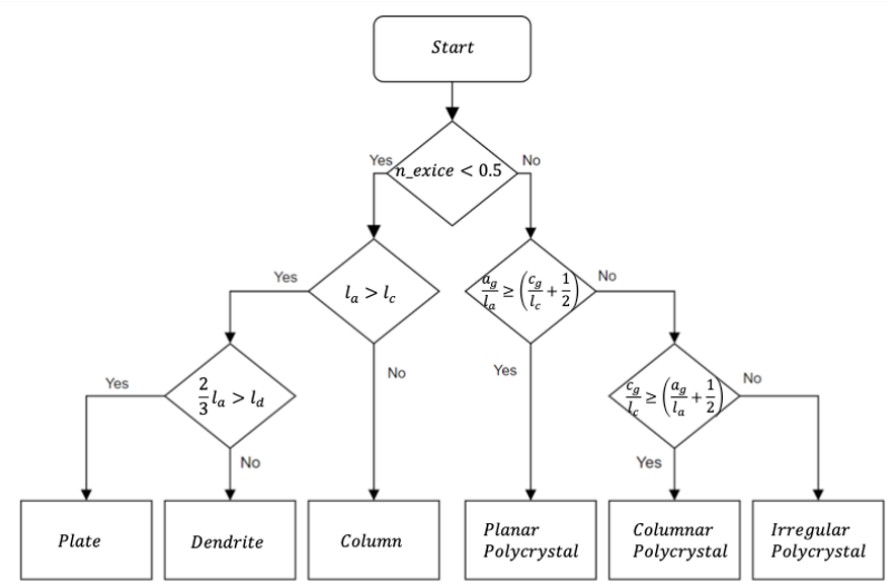

**Figure 2.** Flowchart depicting the diagnostic procedure for identifying ice particle habits in the AMPS.

and Beheng, 2006). Requring assumptions about particle size distribution, mass-size relations, and terminal velocity in PAM-
TRA. In the context of the spectral bin model AMPS, the terminal velocity for the sedimentation of a prognostic variable for
a specific bin is determined by considering the mass, concentration, and the particular type of PPVs within that bin (Hashino
and Tripoli, 2007). Radar reflectivity factor $z_e$ [mm$^6$ m$^{-3}$] is obtained by integrating the normalized particle size distribution
(PSD) $n(D)$ over the entire range of particle sizes D,

$$z_{\mathrm{e}} = \int 10^{18} \sigma_{\mathrm{B}}(D) n(D) \frac{\lambda^4}{\pi^5 |K_{\mathrm{w}}|^2} dD \tag{1}$$

Where $\lambda$ is the wavelength in meters, $|K_{\mathrm{w}}|$ corresponds to the dielectric factor of water, and $\sigma_{\mathrm{B}}(D)$ stands for the backscatter-
ing cross-section of individual hydrometeor particles in square meters (m$^2$). Typically, radar reflectivity is used in logarithmic
units converted with $Z_{\mathrm{e}}[\mathrm{dBZ}] = 10 \log_{10} z_{\mathrm{e}}[\mathrm{mm}^6 \mathrm{m}^{-3}]$.

It is standard practice to use the value for liquid water at centimeter wavelengths ($|K_{\mathrm{w}}| = 0.93$ at Ka-band; Ulaby et al., 1981)
regardless of whether ice or liquid clouds are observed. However, $|K_{\mathrm{w}}|$ is also frequency-dependent. This study employs the
Self-Similar Rayleigh-Gans approximation (SSRGA) parameterization proposed by Hogan et al. (2017) for the backscattering
cross-section $\sigma_{\mathrm{B}}(D)$ calculation. This parameterization is determined by five dimensionless parameters: $\alpha_e$, $K$, $\beta$, $\gamma$, and
$\zeta$. The AR of the particles is represented by $\alpha_e$, whereas $K$ measures the mean mass distribution of the particle along the
propagation direction and is referred to as the kurtosis parameter. The mass fluctuations around the mean mass distribution are
described by $\beta$ and $\gamma$, which represent the power law prefactor and exponent, respectively. $\zeta$ is a correction term for the power

spectrum of the smallest wavenumber. We choose the SSRGA coefficients depending on the normalized particle rime mass fraction following Maherndl et al. (2023).

## 3 Model description and simulation setup

### 3.1 Initial profile

We selected a 1D mixed-phase stratocumulus case for our experiments with the KiD-AMPS framework. To minimize the effects of the microphysics schemes, the temperature field is kept constant. The vertical velocity demonstrates repeated up/down oscillation, and an additional vapor source is supplied to artificially recreate a quasi-steady stratocumulus condition. The vertical velocity is given as

$$
w(z,t) = \begin{cases} w_1 \frac{z}{z_1} \left(1 - \exp\left[-\left(\frac{z-z_1}{z_2}\right)^2\right]\right) \sin\left(\pi t / t_1\right), & \text{if } z < z_1 \\ 0.0, & \text{otherwise} \end{cases}
\tag{2}
$$

and the additional forcing

$$
\frac{dq}{dt}_{\text{add}}(z,t) = \frac{dq}{dt}_{\text{add}}(z,0) = \begin{cases} A\cos\left(\frac{1.25\pi}{2}\right), & \text{if } z < z_3 + z_4, \\ A\cos\left(\frac{z-z_3}{z_4}\frac{\pi}{2}\right), & \text{if } z_3 + z_4 < z < z_3 + 1.25 z_4, \end{cases}
\tag{3}
$$

and A satisfies

$$
\int_0^{z_5} \frac{dq}{dt}_{\text{add}}(z,0) dz = f_{\text{q}}/3600
\tag{4}
$$

The parameters $z_1$, $z_2$, $z_3$, $z_4$, and $z_5$ are set to 450 m, 200 m, 400 m, 100 m, and 1000 m, respectively. Similarly, the values 225 of $\omega_1$, $t_1$, and $f_{\text{q}}$ are set to 1.0 m/s, 600 s, and 5 mm/h, respectively. The initial potential temperature and specific humidity profile are displayed in Figure 3. This implies that the cloud's top region exhibits a temperature of approximately $-20\,°C$, whereas the lowermost part of the cloud shows a temperature of $-15\,°C$. Importantly, the profiles were specifically crafted to emulate a mixed-phase stratocumulus layer, drawing inspiration from the GCSS (Global Energy and Water Cycle Experiment Cloud System Study) SHEBA (Surface Heat Budget of the Arctic Ocean) intercomparison, a choice substantiated by previous 230 studies (Klein et al., 2009; Morrison et al., 2011; Fridlind et al., 2012). This selection was made with the deliberate intent of aligning our study's environmental conditions with well-established benchmarks, thereby enabling seamless comparisons with other investigations centered around similar Arctic settings. The GCSS SHEBA intercomparison is widely acknowledged for providing meticulously documented atmospheric conditions that faithfully represent Arctic stratocumulus clouds, rendering it an ideal resource for our initial profiles.

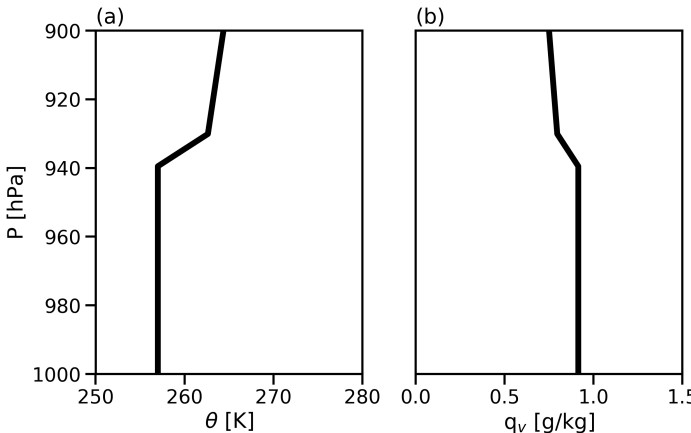

**Figure 3.** The initial condition of the (a) potential temperature $\theta$ [K] and (b) specific humidity $q_v$ [g/kg] profile in KiD-AMPS simulation.

## 3.2 Experimental designs (CCN/INP)

This study aimed to investigate the impact of varying initial concentrations of CCN and INPs on the formation and evolution of clouds. It is widely recognized that aerosols can significantly influence cloud formation and evolution, with their role in these processes depending on factors such as size, composition, and concentration. It should be noted that while aerosols can act as a source of INPs, not all aerosols possess this capability. INPs are particular types of particles capable of initiating ice crystal formation at temperatures below 0°C. The precise composition and physical properties of INPs can considerably vary based on specific conditions, with mineral dust, biological particles (e.g., bacteria and fungi), and certain anthropogenic particles (e.g., industrial emissions) being the most prevalent sources of INPs (Hoose and Möhler, 2012). For this research, we employ Ammonium Sulfate as CCN and assume that the insoluble portion of all aerosols consists of montmorillonite as INP sources. Hashino et al. (2020) demonstrated that the volume-dependent Bigg's immersion method (Diehl and Wurzler, 2004) is to be well-suited for characterizing the ice nucleation process. Consequently, we adopt Bigg's immersion freezing method in this study as well. For further details on ice nucleation schemes, de Boer et al. (2010) (2013) and Hashino et al. (2020) can be referred to.

Within the AMPS model, aerosol particles are represented by lognormal size distributions, which provide comprehensive coverage of their size range. In the CCN activation scheme, the particle size distribution is divided into 10 bins, and for each bin, the critical supersaturation value is individually computed. As the iteration proceeds, bins that exhibit a critical supersaturation below the threshold of environmental supersaturation are identified and subsequently transferred to the liquid spectrum. This methodology ensures the appropriate incorporation of bins in the liquid phase, leading to a faithful depiction of cloud microphysics within the AMPS model. As previously mentioned, montmorillonite particle number concentration serves as an INP proxy for this simulation. For the present study, we conduct sensitivity analyses in this study by altering CCN and

**Table 2.** Definition of the experiments EXP1–5 referred to in the remainder of this study. The simulations are conducted using various concentrations of CCN and INP aerosols.

| | | CCN [cm$^{-3}$] | | | | |
|---|---|---|---|---|---|---|
| | | 10 | 50 | 500 | 1000 | 5000 |
| | 0.001 | X | EXP3 | X | X | X |
| INP [L$^{-1}$] | 0.1 | EXP4 | EXP2 | EXP5 | X | X |
| | 10 | X | EXP1 | X | X | X |

INP proxy concentrations across an extensive range, from extremely low to exceptionally high levels, as illustrated in Table 2. The initial CCN concentrations for these sensitivity simulations are set at 10, 50, 500, 1000, and 5000 cm$^{-3}$ (denoted as CCN10, CCN50, CCN500, CCN1000, and CCN5000, respectively). For each CCN condition, simulations are carried out with initial INP particle concentrations of 0.001, 0.1, and 10 L$^{-1}$, respectively, labeled as INP0.001, INP0.1, and INP10. A similar range for the experimental setup was also selected in the sensitivity study of Fan et al. (2017). The authors further emphasize that in heavily polluted regions like China and India, where CCN concentrations exceeding 1000 cm$^{-3}$ are prevalent, such high values hold significant implications for precipitation extremes and water cycles. Choudhury and Tesche (2023) provides a comprehensive global multiyear dataset of height-resolved concentrations of cloud condensation nuclei (CCN) categorized by aerosol types. These estimates are derived from the spaceborne lidar instrument aboard the Cloud-Aerosol Lidar and Infra-Red Pathfinder Satellite Observation (CALIPSO) satellite. Notably, recent studies demonstrate that the levels of extreme CCN concentrations in heavily polluted regions can exceed 5000 cm$^{-3}$. The selected INP range is well motivated with respect to the variability that can be found in existing in-situ and remote-sensing studies of INP concentrations. Specifically, we based the decision for the lowest value of INP concentration on the values reported for the free troposphere over the Southern Ocean site of Punta Arenas, Chile (Radenz et al., 2021). The selected maximum value of INP concentration of 10 L$-1$ can, e.g., be observed in the case of strong Saharan dust outbreaks in the Mediterranean region as was reported for instance by Ansmann et al. (2019). The range also agrees well with in-situ measurements, as reported by DeMott et al. (2010).

In total, we conducted 15 experiments, and from this set, we specifically selected 5 representative cases (EXP1–5) for detailed analysis in Section 4.1 of the Results (see Table 2). These selected cases encompass clean, pristine, and polluted scenarios, achieved by varying the concentration of CCN and adjusting the concentration of INP to be 100 times smaller and 100 times larger than the commonly observed value of 0.1 L$^{-1}$ in Arctic mixed-phase clouds. The results of the sensitivity test, covering all cases, will be presented in Section 4.2. Additionally, Section 4.3 will showcase the outcomes of coupling a typical EXP1–5 cases with PAMTRA for radar retrieval analysis.

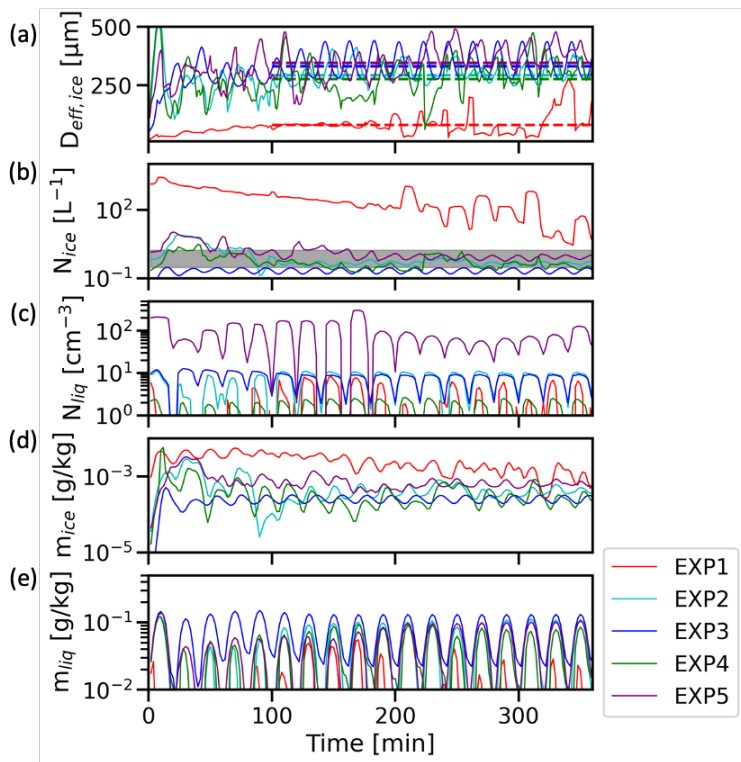

**Figure 4.** Time series of key variables: (a) effective diameter of ice crystals ($D_{eff,\ ice}$) with the dotted line representing the average value from 100 minutes, (b) ice number concentration ($N_{ice}$), (c) liquid number concentration ($N_{liq}$), (d) mixing ratio of ice ($m_{ice}$), and (e) mixing ratio of liquid ($m_{liq}$). The gray-shaded regions correspond to the ranges derived from observational data obtained during the SHEBA campaign

## 4 Results

### 4.1 Comparative analysis of CCN and INP effects on mixed-phase clouds

We analyze the evolution of cloud formations and the progression of hydrometeors by altering the initial CCN and INP concentrations. In this chapter, our primary focus will be on Cases EXP1–5 (see Table 1), which represent the most contrasting experimental scenarios among the 15 cases examined.

Figure 4 presents the temporal evolution of cloud development, specifically focusing on the average column value over time. This figure provides valuable insights into the progression of cloud formation in relation to the initial conditions, considering the variations in CCN and INP. In Figure 4(a), we present the effective diameter values of ice particles obtained from the simulations. In this study, the effective diameter of ice particles, denoted as $D_{\text{eff, ice}}$, is calculated using Equation 5.

$$D_{\text{eff, ice}} = 2\sqrt{V_{\text{cs}}/(\pi l_{c,\text{sm}})}$$ (5)

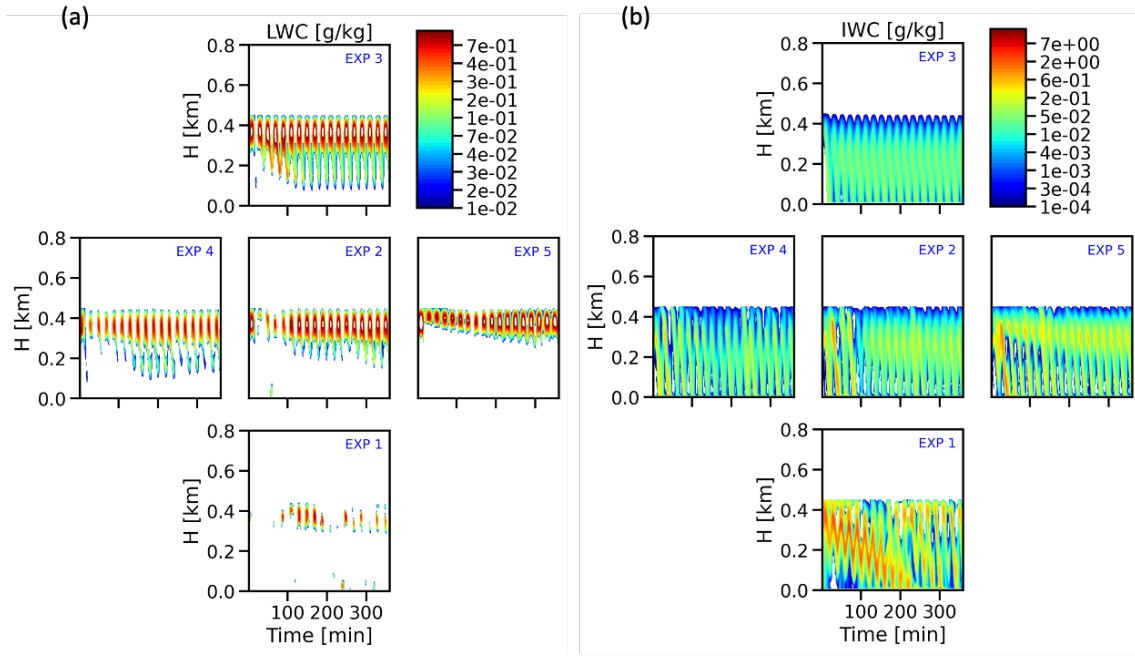

**Figure 5.** Vertical distribution comparison of (a) liquid water content (LWC) [g/kg] and (b) ice water content (IWC) [g/kg] across EXP1–5.

The mean effective diameters after $t = 100$ min for EXP1, EXP2, EXP3, EXP4, and EXP5 are 79 $\mu$m, 293 $\mu$m, 331 $\mu$m, 275 $\mu$m, and 346 $\mu$m, respectively. It is observed that as the concentration of INP increases, there is a corresponding decrease in the effective diameter of ice particles, as evident in EXP1, EXP2, and EXP3. Conversely, an increase in CCN concentration
results in a slight increase in the effective diameter, as observed in EXP4 and EXP5. The simulated effective diameters of ice particles span a wide range, ranging from small values in the tens of microns to larger values in the range of thousands of microns. These findings are consistent with previous observations indicating effective diameters within the range of 300 $\mu$m to 800 $\mu$m (Morrison et al., 2011).

AMPS is comprised of two separate bin spectra. The first 40 bins represent the liquid phase and are categorized as either
cloud or rain, while the next 20 bins represent the ice phase. The mixing ratio and number concentration of liquid and ice particles are found to remain in a quasi-steady state, with the exception of EXP1. Moreover, the experimental results in Figure 4(b) for the number concentration of ice ($N_{ice}$) in EXP2, EXP4, and EXP5, ranging from 0.3 to 1.7 $L^{-1}$ (gray zone), are in agreement with previous observations from SHEBA campaign of mixed-phase clouds (Morrison et al., 2011; Fridlind et al., 2012). These results further suggest that the experiment accurately represents typical mixed-phase clouds. Figure 4 indicates
that as INP concentrations increase in EXP1, EXP2, and EXP3, both $N_{ice}$ and $m_{ice}$ significantly increase, while $N_{liq}$ and $m_{liq}$ decrease. Conversely, the mean $m_{ice}$ is slightly higher in experiments with increased CCN (EXP4, EXP2, and EXP5) despite the higher $N_{liq}$ and $m_{liq}$, consistent with findings from earlier studies employing 2-moment bulk microphysics simulations (Solomon et al., 2018) and experimental investigations (Desai et al., 2019).

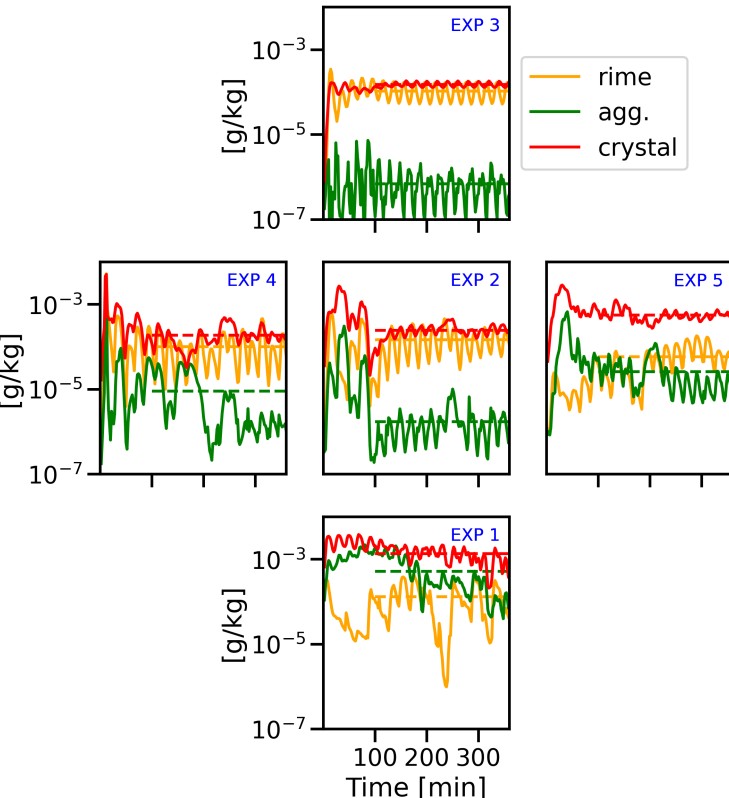

**Figure 6.** Temporal evolution comparison of the mean mixing ratios of ice processes with the dotted line representing the average value from 100 minutes, including melting, riming, aggregation, and crystal across EXP1–5. The aggregation process is abbreviated "agg.".

The impact of CCN and INP perturbations on cloud evolution is illustrated in Figure 5. It is observed that increasing CCN
concentration can significantly affect the relationship between liquid water and ice mixing ratios. In response to the increased concentration of INP, the ice water content increased, as well. In the high-INP scenario (EXP1), the conversion of water droplets into ice crystals occurs more rapidly and efficiently. Consequently, this case predicts almost fully glaciated clouds. When the number of CCN increases, more cloud droplets are formed from within the reservoir of available water vapor. This can lead to an increase in the number concentration of cloud droplets within the cloud as shown in Figure 4(c), which can, in
turn, reduce the size of individual cloud droplets. This reduction in droplet size increases the altitude of the mixed-phase cloud base and reduces the amount of precipitation, which can result in an increase in cloud water mass, given that the cloud is not already saturated to the available water vapor. In summary, the alteration of cloud particle concentration due to perturbations in the aerosol concentrations leads to adjustments in cloud and precipitation patterns, even in the absence of cloud-dynamics interaction, as previously observed in studies by Seifert et al. (2012), Boucher et al. (2013), Heyn et al. (2017), Possner et al.
(2017), Solomon et al. (2018), and Zhang et al. (2018).

Next, we will investigate the response of the ice-phase processes of AMPS to the aerosol perturbations. In the AMPS model, the mass of an ice crystal is partitioned into various process-oriented categories, including pristine crystal mass, aggregated mass, riming mass, and melted water mass, which are tracked as PPVs. The bin components are created through microphysical processes that act upon them, including vapor deposition onto ice crystals, which produce ice crystal mass, melting processes which produce liquid mass, aggregation processes which generate aggregate mass, and riming processes which produce rime mass. The sum of these four components makes up the total mass. In Figure 6, the mixing ratio of ice water content is presented separately for the 4 ice-phase processes and for each of the 5 selected experiments. High concentrations of INPs lead to an increased number concentration of ice particles (see Figure 4) and consequently to increased ice water mass, as the presence of INPs allows the formation of ice crystals to occur more readily, resulting in an increased number of collisions and coalescence among the ice crystals. The results demonstrate that as the number of aerosols and concentration of INPs increase, there is a corresponding increase in the frequency of aggregation due to the higher concentration of ice crystals that promote this process. Conversely, it is observed that the occurrence of riming is reduced when there is a shortage of sustained supercooled liquid layers.

The concentration of CCN directly impacts the condensation of water vapor into liquid droplets. With an increase in CCN concentration, a greater number of smaller liquid droplets is formed. However, these smaller droplets have lower inertia and are less effective at colliding with ice particles. As a result, the efficiency of the riming process decreases as the number of small supercooled droplets increases. This observation aligns with a previous observation study conducted by Borys et al. (2003). Additionally, the persistence of smaller droplets that freeze before colliding with ice particles and before being able to contribute to riming can grow via the Bergeron-Findeisen process (Bergeron 1935; Findeisen 1938). This process, which operates most efficiently within the temperature range of $-15$ to $-20\,°C$, occurs in mixed-phase clouds where both supercooled water droplets and ice particles coexist. In the Bergeron-Findeisen process, water vapor tends to preferentially condense onto the ice particles due to the lower saturation vapor pressure over ice compared to water. As a result, the preferential growth of ice particles leads to an increase in the number of ice particles and a decrease in riming mass. Figure 4 and Figure 5 illustrate how elevated CCN concentrations lead to an increase in the number of ice particles, specifically EXP4 and EXP5, and a decrease in riming efficiency.

Therefore, fluctuations in CCN and INP concentrations have significant effects on precipitation formation by modifying the amount of supercooled liquid water. The results indicate that polluted clouds exhibit a reduced size and higher amounts of supercooled liquid water, resulting in precipitation preferably formed through ice phase aggregation rather than riming processes. Conversely, pristine clouds characterized by a larger size and lower amounts of supercooled liquid water predominantly generate precipitation through riming processes.

The frequency of ice particle freezing is influenced by the sizes of liquid particles that freeze most frequently in each scenario. The parameterization of immersion freezing takes into account the volume of the droplets, indicating that larger droplets have a higher probability of freezing. However, the number concentration of the droplets also plays a role, with smaller droplets being more prevalent. Consequently, drawing definitive conclusions about the overall effect becomes challenging. Figure 7 illustrates the size distribution of ice particles based on number weighting. The results demonstrate that an increased concentration of

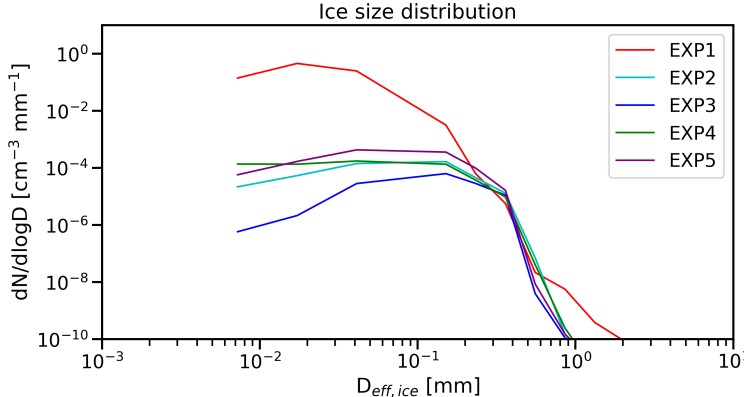

**Figure 7.** The mean ice particle size distribution averaged after $t = 100$ min.

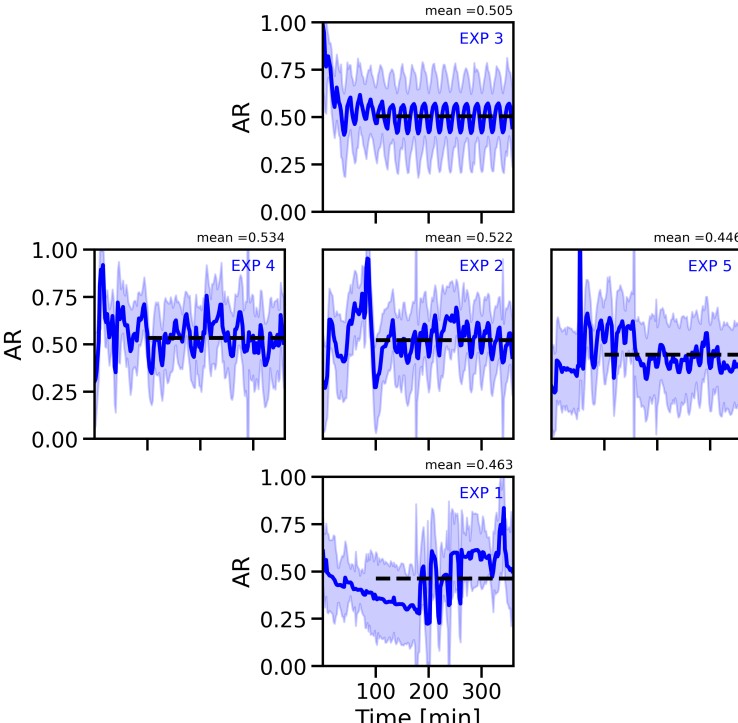

**Figure 8.** Comparison of the temporal evolution of the mean aspect ratio (AR) with the black dotted line representing the average value from 100 minutes across EXP1–5.

INPs corresponds to a significant abundance of small ice particles. When comparing EXP1 and EXP2, it's worth highlighting that despite EXP1 showing a smaller average size, it still exhibits a substantial population of small particles. This is further

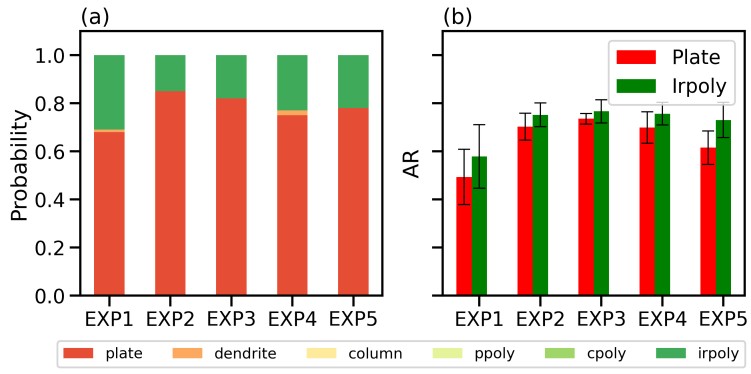

**Figure 9.** Comparative analysis of the probability of (a) ice habit classification, including plate, dendrite, column, planar polycrystal (pploy), columnar polycrystal (cpoly), and irregular polycrystals (irpoly), and (b) mean AR and standard deviation (with error bars) for plate and irregular polycrystals during the two-hour period following 100 minutes across EXP1–5.

compounded by the aggregation process, which plays a pivotal role in transforming these initially smaller hydrometeors into relatively larger ice particles.

In this study, Figure 8 presents the correlation between the number-weighted AR of ice particles and the concentrations of INPs and CCN in mixed-phase clouds. Notably, the results indicate that regardless of the conditions, the ice particles exhibit an oblate-like shape with a specific AR within the temperature range from $-20$ to $-16\,°C$. Furthermore, the AR decreases as the INP concentration increases. This is attributed to the fact that higher INP concentrations promote the formation of smaller and more compact ice crystals with extreme ARs, in contrast to larger crystals that develop in low INP environments. The

smaller-sized crystals exhibit a more pronounced AR due to the non-linear feedback ($\frac{dl_c}{dl_a} = \Gamma(T)l_c/l_a$), where $\Gamma$ represents the inherent growth ratio of ice crystals, reflecting the current atmospheric conditions and the change in the mass of the ice particle (Chen and Lamb, 1994; Hashino and Tripoli, 2007; Hashino et al., 2020).

Similarly, in the scenario with low CCN concentration (EXP4), ice particles undergo growth through the deposition and riming process, resulting in more spherical shapes. Conversely, in the high CCN scenario, the freezing of droplets through

the immersion mode occurs at smaller sizes, leading to the nucleation of smaller ice crystals and the subsequent formation of smaller ice particles. When comparing EXP4 and EXP5 in Figure 7, an increase in CCN concentration is observed to coincide with a significant rise in the concentration of smaller ice particles, accompanied by a decrease in the abundance of larger ice particles around 1 mm. Consequently, the overall average value exhibits a decreasing trend with increasing CCN concentration, primarily driven by the greater number of small ice particles. This phenomenon also leads to an increase in ice water mass

and the availability of water vapor for ice particle growth, resulting in a higher abundance of small ice particles and a slight decrease in the AR, as illustrated in Figure 8.

To gain a better understanding of AR, we examined AR with each ice habit in detail. As shown in Figure 9, the ice particles in the mixed-phase clouds, where the temperature ranges from $-20$ to $-16\,°C$, primarily exhibit an oblate-like shape with AR ($\alpha$) $< 1$, consisting of plates and irregular polycrystal habits. The result agrees with earlier findings from laboratory studies (Bailey

and Hallett, 2009) and observations (Auer and Veal, 1970). Most droplets that freeze close to $-20\,°C$ are polycrystalline in nature, as observed in polycrystalline ice habit frequency from in situ observations (Bailey and Hallett, 2009).

To clarify, in the AMPS model, if the maximum dimension of the ice crystal is less than $20\,\mu m$ and the temperature is below $-20\,°C$, the growth regime is determined using the corresponding cumulative relative frequency of habits and a random number generator. In this case, small ice crystals can exhibit growth modes that differ from the habit diagnosed at the start of the time
step. However, for temperatures above $-20\,°C$, it is assumed that polycrystals do not form. Once the maximum dimension of the ice crystal exceeds $20\,\mu m$, it is assumed to follow the growth of the diagnosed habit at the beginning of the time step. This indicates that most irregular polycrystals observed in this experiment primarily formed within the initial cloud top layer, where temperatures are near $-20,°C$. The result is consistent with the findings of Hashino et al. (2020).

Figure 9(a) provides insights into the predominant shape of ice particles, with plates representing the majority of the particles.
EXP1 exhibits a plate fraction of 68%, while the remaining experiments range from 75% to 85%. As the concentration of INPs increases, a greater number of ice particles form, as shown in Figure 4. Over time, the initially formed plate-shaped ice crystals undergo a transformation influenced by the aggregation process, resulting in the emergence of irregular polycrystals. Notably, the AR of plates gradually decreases compared to that of irregular polycrystals as the INP concentration increases. In EXP1, as depicted in Figure 8, the AR demonstrates an increase after 200 minutes, primarily attributed to the formation of irregular
polycrystals. This phenomenon arises from the aggregation process involving newly formed small ice particles and pre-existing ice particles. The resulting irregular polycrystals exhibit a higher AR compared to the plate-shaped crystals, contributing to an overall increase in the average AR. In EXP1, the aggregation process becomes more pronounced, leading to the formation of fully glaciated clouds and a notable increase in the occurrence of irregular polycrystals. A comparison between EXP4 and EXP5 in Figure 9(a) reveals similar proportions of the plate and irregular polycrystal shapes. However, Figure 9(b) shows that
the AR is smaller in EXP5. This observation is consistent with previous results, indicating that higher concentrations of CCN lead to higher ice particle concentrations, resulting in a lower AR.

In conclusion, Figure 9 provides compelling evidence of a correlation between the concentration of INPs, CCN, and the AR of ice particles. The findings indicate a decrease in the AR as the concentrations of INPs and CCN increase.

## 4.2 Sensitivity test

In Section 4.1 of this study, the primary focus is the analysis and comparison of the five most frequently observed scenarios, exploring the changes observed over the entire duration of the experiment. Now, in Section 4.2, the focus shifts to examining the average values, specifically excluding the initial 100 minutes for each experiment. The objective of this section is to investigate the tendencies and influences of individual ice microphysical processes, as well as to assess the impacts of varying dynamics and initial INP types across a total of 15 setups.

Figure 10 illustrates the mean freezing rate of ice particles under different CCN and INP scenarios. In the AMPS model, the source terms of ice mass include ice nucleation, vapor deposition, aggregation, riming, hydrodynamic breakup, and melting-shedding. In mixed-phase clouds, ice particles primarily grow through vapor deposition, a process in which water vapor molecules freeze onto their surface. According to the experiments, deposition is the primary process for ice growth, further

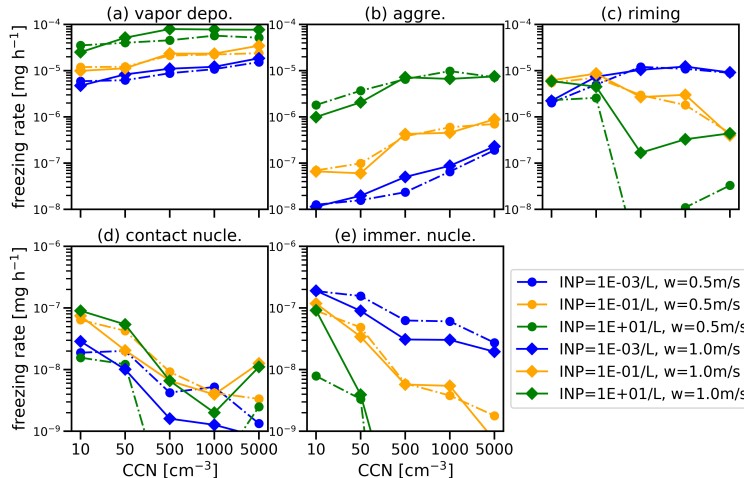

**Figure 10.** Comparison of mean freezing process rates, including (a) vapor deposition, (b) aggregation, (c) riming, (d) contact nucleation, and (e) immersion nucleation across 15 cases. The model outputs for the process rates are recorded every 1 minute, and the data are spatially averaged over the domain grid points, excluding the first 100 minutes. The dashed line represents the value of $w_1$ when we decrease from $1.0\,\mathrm{m\,s^{-1}}$ to $0.5\,\mathrm{m\,s^{-1}}$, as defined in Eq. (2).

enhanced by increasing CCN and INP concentrations. The riming and aggregation processes are the next significant contribu-
tors, followed by the nucleation process. The hydrodynamic breakup and melting-shedding processes are not activated in this
experiment.

The increase in INP concentration amplifies the contact nucleation process, aggregation process, and subsequent deposi-
tional growth, while the riming-related growth and immersion nucleation process show a decline. At low INP concentrations,
the riming-related growth is comparable to deposition. However, as the INP concentration rises, the contribution of riming
considerably diminishes, as discussed earlier. Consequently, the availability of liquid becomes a limiting factor for riming.
The contribution of riming exhibits a significant decrease with increasing CCN concentration, except in low INP conditions.
Notably, when a substantial difference exists, the difference can exceed 10 times the values observed in between different
scenarios. The major microphysical processes, including vapor deposition, riming, and nucleation, exhibit high sensitivity to
INPs. Their sensitivity becomes more significant in high INP conditions, accompanied by significant changes in dynamics
and thermodynamics. To investigate the impact of dynamic factors, we decreased $w_1$ from $1.0\,\mathrm{m\,s^{-1}}$ to $0.5\,\mathrm{m\,s^{-1}}$ in Eq. (2),
leveraging the capability of KiD-AMPS to differentiate between dynamic and microphysical effects. The results confirm that
an increase in $w_1$ amplifies the vapor deposition rate, aggregation rate, and contact nucleation processes while reducing the
immersion freezing rate. However, the overall trends exhibit a similar pattern.

According to Hashino et al. (2020), the volume-dependent Bigg's immersion method Diehl and Wurzler (2004) is a suitable
approach for modeling mixed-phase clouds (see Eq. (1) in Hashino et al. (2020)). To account for this information, we adapt
Bigg's immersion freezing scheme for our study. In this scheme, the freezing rate is strongly influenced by the droplet mass

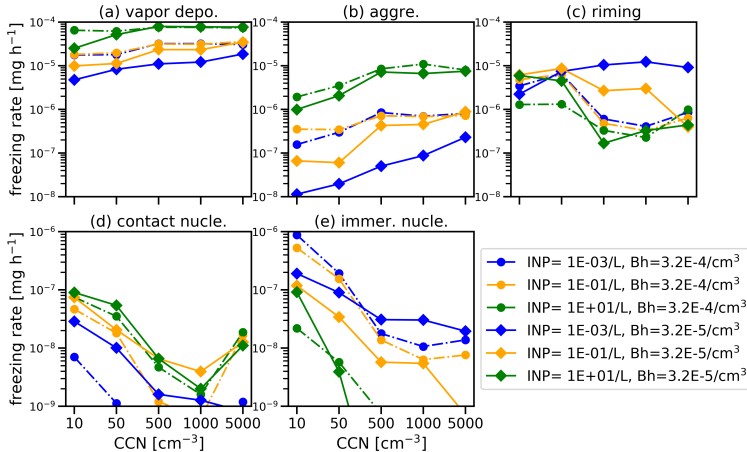

**Figure 11.** Comparison of the mean freezing process rates, including vapor deposition, riming, contact nucleation, and immersion nucleation, across 15 cases. The data presented in the plots were processed using the same methodology as described in Figure 10. The dashed line in the plots represents the enhanced value of the immersion freezing rate ($Bh = 3.2 \times 10^{-4}$ cm$^{-3}$) as defined in Eq. (1) in the study by Hashino et al. (2020).

as well as Bh and Temperature. Bh represents the freezing efficiency of insoluble material in the droplet and is currently set to $3.2 \times 10^{-5}$cm$^{-3}$ for montmorillonite, as reported by Diehl and Wurzler (2004). To explore the effects of varying Bh, we multiplied the parameter by a factor of 10 and compared the resulting changes in freezing rates. The findings are presented in

Figure 11, indicating that an increase in Bh results in an increase in immersion freezing rate, aggregation production rate, and a minor increase in vapor deposition rate, while the effect of contact nucleation decreases. Nonetheless, the vapor deposition process remains the dominant factor, impacting the AR by reducing it as Bh increases. This is similar to the effect of INP increase, which generates smaller AR. The prominence of this effect can be attributed to the enhanced production of small droplets, as depicted in Figure 7.

The relationship between CCN concentration and the rate of ice production through immersion freezing in mixed-phase clouds is complex and depends on multiple factors. In theory, an increase in CCN concentration can result in the formation of more cloud droplets. If these droplets become supercooled, they can potentially serve as sites for ice formation through immersion freezing when INPs are present. This suggests a potential increase in the rate of ice production with higher CCN concentration. However, the results reveal a decrease in the contribution of immersion nucleation with increasing CCN concen-

tration. This can be attributed to the twofold effect of increasing CCN. On the one hand, it can lead to a higher number of cloud droplets available for freezing. On the other hand, it tends to produce smaller cloud droplets due to increased competition for available water vapor, known as the Twomey effect (Twomey, 1974). Smaller droplets require colder temperatures to freeze, making them less likely to initiate ice formation. As a result, an increase in CCN concentration does not lead to a corresponding increase in the rate of ice production if the droplets are too small to freeze. Moreover, as the concentration of INP increases,

the mass contribution of the immersion freezing rate also decreases. As depicted in Figure 5, an increase in INP concentration

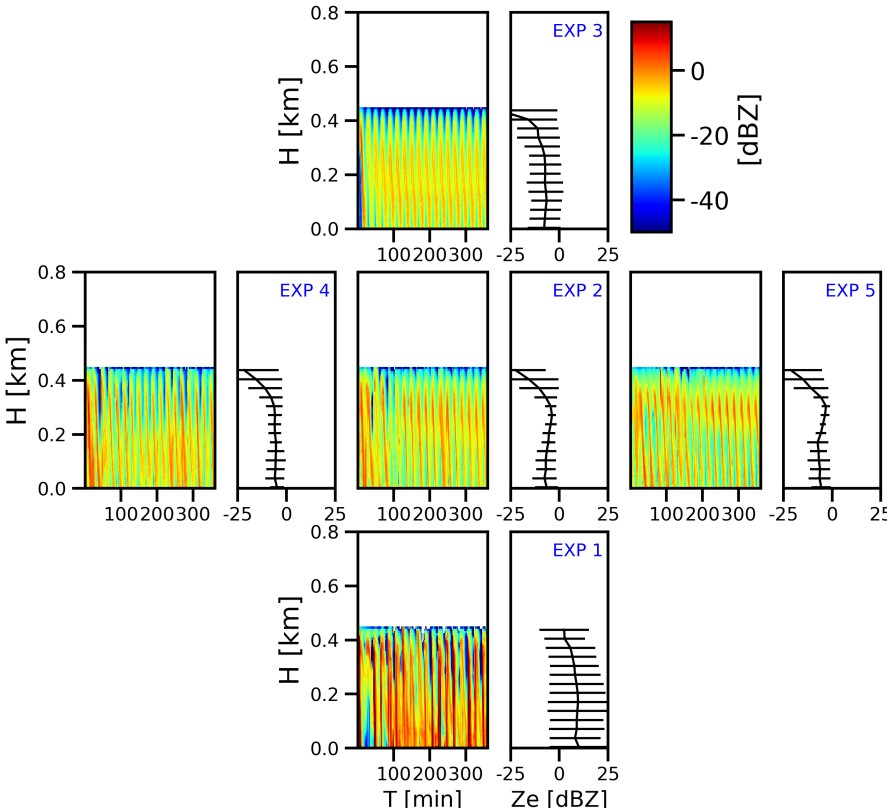

**Figure 12.** Comparison of the distribution and mean vertical profile of forward-modeled radar reflectivity factor $Z_e$ [dBZ] at Ka-band using the PAMTRA across EXP1–5.

leads to a decrease in LWC, as well as a reduction in the areas of ice particles and coexistence. The immersion freezing rate is influenced by the average mass of droplets in the same mass bin and location. Therefore, it is evident that the immersion freezing rate also declines with increasing INP concentrations.

To sum up, the relationship between aerosol concentration and the rate of ice production through immersion freezing is intricate and influenced by various factors, including the properties and concentration of CCN and INPs, temperature, and supersaturation level within the cloud. To definitively determine the actual effect of increasing INP and CCN on the rate of ice production through contact freezing and immersion freezing, careful cloud microphysical modeling or observations are necessary.

### 4.3 PAMTRA coupled with AMPS

In this final result section, the application of PAMTRA to the simulations EXP1–5 will be elaborated to demonstrate the sensitivity of radar observations to changes in CCN and INP.

As radar reflectivity is determined by multiple parameters, including size, density, and AR (see Eq. (1)), forward simulations are required in order to evaluate the effects of INP and CCN perturbations on observable parameters such as the radar reflectivity. The results of the forward simulations for the 5 experiments, as explained in Sect. 2.3, are shown in Figure 12. The spectral-bin model reveals a clear decrease in particle size as aerosol loads increase, which leads to a decrease in radar reflectivity due to the strong dependence of backscattering on particle size to the power of 6. However, as shown in Figure 12, the radar reflectivity increases with increasing INP concentrations, which is consistent with observations (Zhang et al., 2018; Radenz et al., 2021) . The increase in reflectivity is attributed not only to changes in particle size but also to alterations in number concentration, AR, and other parameters.

In the context of analyzing radar reflectivity data, it is essential to determine the mean vertical profile in order to understand the atmospheric structure and characteristics at various altitudes. This information provides valuable insights into the vertical distribution of hydrometeors. Examining the profiles in Figure 12, we observe a noticeable increase in reflectivity at the cloud bottom (at around 300 m) as the CCN concentration rises. This increase can be attributed to the greater number of cloud droplets resulting from the higher CCN concentration. It is noteworthy that EXP5 displays a remarkably low standard deviation in the cloud bottom at 300 m, indicating a relatively uniform distribution of particle sizes in that region. Consistent with expectations, an increase in the INP concentration leads to elevated reflectivity across different altitude levels, accompanied by an increased standard deviation. Higher mean reflectivity values in EXP1 show a larger abundance of hydrometeors in the observed scenarios. However, an intriguing finding is the contrasting pattern observed in the cloud top, where the standard deviation demonstrates a decrease.

**Table 3.** Statistical analysis of the radar reflectivity factor ($Z_{\mathrm{e}}$) [dBZ] distribution, which includes the computation of mean ($\mu$), standard deviation ($\sigma$), and skewness ($\tilde{M}$) values. The data are spatially averaged over the grid points of the domain, excluding the initial 100 minutes.

| | | CCN [$cm^{-3}$] | | |
|---|---|---|---|---|
| ($\mu, \sigma, \tilde{M}$) | | 10 | 50 | 500 |
| | 0.001 | | EXP3 (-11.83, 9.12, 0.36) | |
| INP [$L^{-1}$] | 0.1 | EXP4 (-10.23, 5.70, 1.63) | EXP2 (-10.09, 5.60, 1.19) | EXP5 (-9.87, 5.05, 1.71) |
| | 10 | | EXP1 ( 4.65, 12.47, 2.87) | |

Table 3 presents the comprehensive statistics of reflectivity, detailing the total (temporal and vertical) mean ($\mu$), standard deviation ($\sigma$), and skewness ($\tilde{M}$). Our examination of radar reflectivity data unveils distinct statistical patterns that are influenced by the concentrations of CCN and INP. For instance, we find that the mean radar reflectivity is $-10.09$ dBZ, accompanied by a standard deviation of 5.60 dBZ and a skewness of 1.19 for the reference mixed-phase stratocumulus cloud case in EXP2. These values indicate a relatively symmetrical distribution with a slight rightward skew. As the CCN concentration increases while maintaining the same INP level in EXP5, we observe a marginal rise in the mean reflectivity to $-9.87$ dBZ, a decrease in the standard deviation to 5.0 dBZ, and an increase in the skewness to 1.71, suggesting a more pronounced skewness in the distribution.

On the other hand, where the INP concentration is elevated to $10\,\mathrm{L}^{-1}$ at a CCN concentration of $50\,\mathrm{cm}^{-3}$ in EXP1, we note a significant increase in the mean reflectivity to $4.65\,\mathrm{dBZ}$, a larger standard deviation of $12.47\,\mathrm{dBZ}$, and a substantially higher skewness of 2.87. These findings signify a more dispersed and highly skewed distribution, possibly indicating the presence of different hydrometeor types and size variations, including precipitation particles. In this study, this is linked to the aggregation of ice particles. Hydrometeors are plate and irregular polycrystals, most of which are plates. However, as depicted in Figure 7, the size distribution of ice particles is quite diverse in EXP1. Conversely, when the INP concentration is very low at $0.001\,\mathrm{L}^{-1}$ while maintaining a CCN concentration of $50\,\mathrm{cm}^{-3}$, the mean radar reflectivity measures $-11.83\,\mathrm{dBZ}$, the standard deviation is $9.12\,\mathrm{dBZ}$, and the skewness is 0.36, implying a homogeneous distribution with a slight leftward skew and a low number of ice and liquid particles. Comparing the mean Z values between EXP2 and EXP3 shows minimal differences. However, distinct variations in distribution are observed based on the values of $\sigma$ and $\tilde{M}$.

## 5  Summary and conclusions

This study examines how perturbations in CCN and INP concentrations affect the shape of ice particles in mixed-phase clouds. The results reveal that both CCN and INP concentrations play a vital role in determining the shape of ice particles and influencing cloud microphysics. The effective diameter of ice particles, which indicates their size, is found to be influenced by the concentrations of CCN and INP. Higher INP concentrations result in smaller effective diameters, while increased CCN concentrations lead to a slight increase in size. The size distribution of ice particles ranges from tens of microns to thousands of microns, consistent with previous observations. Analysis of the ice particle shapes shows that oblate-like crystal shapes are most common in the temperature range from $-20$ to $-16\,°\mathrm{C}$. However, a significant presence of irregular polycrystals is observed, especially in scenarios with high INP concentrations. In scenarios with high INP concentrations, we observe a complex interplay of nucleation, aggregation, and collision phenomena. These occur against a backdrop of dynamic changes in cloud stability and limited LWC. Collectively, these processes lead to a broader distribution of ice particle sizes, ranging from larger to smaller, a pattern that is corroborated by our experimental data. The AR of ice particles is affected by both CCN and INP concentrations, with higher concentrations leading to smaller AR values. This indicates that the concentrations of CCN and INP have an impact on the growth and shape of ice particles in mixed-phase clouds. The relationship between CCN concentration and ice particle shape is more complex than that of INP. Increased CCN concentrations promote the formation of more cloud droplets but result in smaller droplet sizes. As a result, the efficiency of the riming process, where supercooled droplets collide with ice particles, decreases with higher CCN concentrations. In conclusion, the concentrations of CCN and INP significantly influence the shape and morphology of ice particles in mixed-phase clouds. Higher INP concentrations lead to smaller and more compact ice particles, while increased CCN concentrations result in a decrease in the AR of ice particles. These findings highlight the intricate connection between aerosol concentrations, microphysical processes, and ice particle shapes in cloud systems. Accurate modeling and prediction of cloud behavior and precipitation require a comprehensive understanding of these relationships.

In our study, we employed the radar forward simulator PAMTRA to analyze simulation results and derive radar variables, uncovering significant disparities in the findings. When INP concentrations are at low to moderate levels, the mean value of $Z_e$ remains relatively consistent, indicating a stable trend. However, distinct variations are observed in the $Z_e$ distributions characterized by $\mu, \sigma$, and skew $\tilde{M}$. Therefore, for future observational studies investigating the impact of aerosols on mixed-phase clouds, it is important to consider incorporating $Z_e$ variability, under the condition that the actual cloud conditions align with the assumptions made in our model. In scenarios with substantial differences in INP concentrations, we observed discrepancies of 5.5 dB. This finding aligns with the results reported by Radenz et al. (2021), where a difference of 5–10 dB was observed between continental (Leipzig) and pristine (Punta Arenas) locations within specific temperature ranges. Furthermore, their study reveals that the disparity in INP concentrations is at least one order of magnitude higher and has the potential to be even greater if the contribution of continental aerosols in Punta Arenas is lower than the maximum assumed value.

The formation of ice crystals with different ARs has been linked to variations in CCN and INP concentrations. However, it is important to acknowledge that factors such as temperature, humidity, and wind shear can also have a significant influence on ice crystal ARs (Barrett and Hoose, 2023). Additionally, the choice of modeling schemes can introduce sensitivity to these ratios. Interestingly, our findings show a decrease in AR with increasing INP concentration, which contradicts the results of a previous study by Ong et al. (2022) where an increase in Bh led to an increase in AR. Ong et al. (2022) also discussed the formation of columnar crystals with an AR greater than 1, which they attributed to slight variations in temperature conditions. Consequently, the AR varied as the INP concentration increased. Moreover, the occurrence of irregular polycrystals at approximately the 200-minute mark highlights the intricate interplay between the initial conditions, dynamics, and microphysics of the simulation in the High INP case. This particular timing, which is unique to our experiment, demonstrates the susceptibility of ice particle formation processes to different conditions. It is noteworthy that, given the consistent vertical wind and humidity conditions established in our simulation, extending the duration would likely result in the persistence of observed cyclical AR patterns, rather than a stabilization of AR values. The complex relationship between ice particle shape and aerosol load is still not fully understood, and further research is needed to unravel the underlying mechanisms governing ice crystal formation and ARs in different environmental contexts.

The research employed the KiD model to investigate cloud microphysics processes, but it is important to consider the limitations of this one-dimensional framework. The simplified nature of the model may not fully capture the complexities and interactions observed in real three-dimensional cloud systems. Additionally, focusing solely on individual microphysical processes within the model might overlook potential uncertainties that arise from their interactions within larger-scale weather models. It is crucial to interpret the findings of this study while recognizing the limitations of the models used, including the omission of key processes such as radiation. The KiD model was developed to understand microphysics schemes without the inclusion of dynamic or radiative feedbacks, which play a significant role in cloud layer heating, cooling, and their impact on cloud microphysics and dynamics. Neglecting these processes could lead to inaccuracies in the representing of cloud temperature, water vapor distribution, and vertical motions within mixed-phase clouds.

To address these limitations, future research will explore shape-integrated simulation models AMPS coupled with 3D dynamic cores such as the ICOsahedral Non-hydrostatic (ICON; Zängl et al., 2015) modelling framework. The simulation data

can then be compared to observational radar data using a radar-forward simulator, selecting an observation dataset from field experiments. By incorporating more sophisticated tools and remote sensing techniques, a more comprehensive and accurate analysis of cloud behavior can be achieved. The aim is to enhance cloud simulations by incorporating these processes to improve realism and accuracy. Furthermore, there is a pressing need for future experiments that address a more detailed distribution of INP and CCN perturbations. Such studies aim to deepen our understanding of the dynamics of aerosol-cloud interactions. This entails evaluating how aerosol perturbations influence the evolution of idealized stratiform mixed-phase clouds, a necessary step before a precise general assessment of aerosol-cloud interactions can be realized. Such efforts will be instrumental in improving the realism and accuracy of cloud simulations by incorporating sophisticated modeling and remote sensing techniques, thereby enhancing our ability to predict cloud behavior and its implications on the climate accurately.

*Code availability.* The Kinematic Driver (KiD) model code is available under the link https://github.com/Adehill/KiD-A.git. The simulation results are available under the link https://doi.org/10.5281/zenodo.8257078 (Lee et al., 2023).

*Author contributions.* JL conducted all simulations, analyzed the data, and drafted the manuscript. JL, PS, and OK contributed initial ideas and designed experiments. PS and TH supervised the work and revised the manuscript. TH provided the AMPS and MM provided the code PAMTRA, and they provided support throughout the work. All authors actively collaborated in the development of the paper and participated in scientific discussions.

*Competing interests.* The contact author has declared that neither they nor their co-authors have any competing interests.

*Acknowledgements.* The work leading to this article was funded by the Deutsche Forschungsgemeinschaft (DFG – German Research Foundation) project SPOMC (project number 408027490) which is part of the DFG priority program SPP 2115 PROM (project number 359922472). We acknowledge the provision of additional funding by SPP 2115 PROM for a research stay of JL in Japan, which enabled the initial cooperation with TH, the key developer of AMPS, for the provision of training in using AMPS. TH was supported by the Japan Society for the Promotion of Science Grant-in-Aid for Scientific Research (C) Grant JP21K03665.

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
