# Peer review of "Simulations of the impact of CCN and INP perturbations on the microphysics and radar reflectivity factor of stratiform mixed-phase clouds"

_EGUsphere, 2023_

## Author Comment (AC1)

**Response to reviewer 1#**

Numerical evidence that the impact of CCN and INP concentrations on mixed-phase clouds is observable with cloud radars

By Junghwa Lee et al.

The authors present a model-based analysis of how changes in CCN and INP would affect mixed-phase cloud properties. The most interesting part of the study is the analysis of the impact of CCN and INP on ice particle properties. It is show that changes in CCN and INP could result in detectable changes in particle shapes, for example. Overall, the article is well-written, but there are a few issues that I have summarised below. My major concern is the analysis of how aspect ratio of particle changes as a function of changes in INP and CCN concentrations. I think the analysis would be clearer is a consistent particle classification would be used throughout the study.

**RC1-Comment 1:**

Page 4. What are particle property variables (PPVs)? It would be helpful to see a list all PPV used in the study.

Thank you for your inquiry about Particle Property Variables (PPVs) mentioned on Page 4 of our manuscript. We appreciate the opportunity to clarify this concept, which is crucial for understanding the particle characteristics in our study.

PPVs, as outlined in our study, are a set of prognostic variables that represent the physical characteristics of particles within a given bin of a cloud microphysical model. These variables are essential in defining the unique attributes of ice particles, particularly in complex atmospheric conditions and within the intricate flow fields of mixed-phase clouds. The PPVs in our study include mass content components, length variable components, volume variable components, and aerosol mass content components.

Each PPV is predicted based on the current atmospheric conditions and the growth history of particles, allowing for a dynamic and accurate rcepresentation of particle properties. This approach, as detailed in Ong et al., 2022 eliminates the need for assuming an analytical distribution for the entire spectrum of particle properties, thus enabling a more realistic simulation of ice nucleation, vapor deposition processes, and the development of ice crystal habits.

Since Hashino explained in detail in the last model development paper, it was not covered in detail in our paper. For a more detailed description of the model, refer to the following previous paper by Ong et al., 2022 in Table 1 and also the following table:

| Spectrum | PPV               | Description                                           |  |
|----------|-------------------|-------------------------------------------------------|--|
| Liquid   | $ ho_{lat}/ ho_m$ | Mixing ratio of total aerosol mass                    |  |
|          | $ ho_{las}/ ho_m$ | Mixing ratio of soluble aerosol mass                  |  |
| Ice      | $ ho_{cry}/ ho_m$ | Mixing ratio of crystal mass                          |  |
|          | $ ho_{rim}/ ho_m$ | Mixing ratio of riming mass                           |  |
|          | $ ho_{agg}/ ho_m$ | Mixing ratio of aggregate mass                        |  |
|          | $ ho_{frz}/ ho_m$ | Mixing ratio of frozen mass                           |  |
|          | $ ho_{mlt}/ ho_m$ | Mixing ratio of meltwater mass                        |  |
|          | $ ho_{iat}/ ho_m$ | Mixing ratio of total aerosol mass                    |  |
|          | $ ho_{ias}/ ho_m$ | Mixing ratio of soluble aerosol mass                  |  |
|          | $n_{ m exice}$    | Extra crystalline structure number                    |  |
|          | $V_{ m cs}$       | Circumscribing volume                                 |  |
|          | $l_{\rm a}^3$     | Cube of the a-axis length                             |  |
|          | $l_{\rm c}^3$     | Cube of the c-axis length                             |  |
|          | $l_{\rm d}^3$     | Cube of the d-axis length (Cube of the dendritic arm) |  |
|          | $a_{g}$           | Center of gravity along the a-axis                    |  |
|          | $c_{g}$           | Center of gravity along the c-axis                    |  |

Table 1. The list shows the 16 advected prognostic variables representing the particle property variables (PPVs) within a liquid and ice spectra across each bin, as utilized in the KiD-AMPS model. Here,  $\rho_m$  denotes the moist air density, while  $\rho_{lat}$  and  $\rho_{iat}$  specify the total aerosol density within the liquid phase and ice phase, respectively. Additionally,  $\rho_{las}$  and  $\rho_{ias}$  correspond to the soluble aerosol density in the liquid and ice phases.

We however agree to the suggestion of the reviewer and included a detailed list and description of all PPVs usind in our study into the revised manuscript. The information can be found in lines 121 and table 1 of the revised manuscript.

**RC1-Comment 2:**

Page 5. The definition of the sphere volume circumscribing the ice particle is not very clear. Do you have a reference or more detailed explanation of how it is defined? It is no clear what the equation on line 137 implies in terms of an assumed particle shape.

Thank you for your comment regarding the definition of the sphere volume circumscribing the ice particle, as mentioned on Page 5 of our manuscript. We appreciate your request for further clarification and the opportunity to enhance the comprehensibility of this important concept in our study.

In our study, the concept of the circumscribing sphere volume is integral to understanding the microphysical processes of ice particles in mixed-phase clouds. As explained in Hashino and Tripoli (2011a), the sphere volume circumscribing the ice particle is a crucial parameter for predicting the mass-dimension (m-D) relationship in AMPS model. This concept is based on functions that connect the predicted circumscribing sphere volume with diagnosed aspect ratio semiaxis lengths and the maximum dimension of the ice particle model. The sphere volume circumscribing the conceptual shape for each mass bin is considered the particle property variable (PPV) most relevant to these predictions. In addition, this is the key factor that determines the effective parameter in future analysis.

This approach assumes a geometrically consistent formulation among the microphysical processes, where the concentration-weighted volume of a circumscribing sphere is transferred among mass bins according to the collection process. This allows for obtaining the circumscribing sphere volume of the representative hydrometeor, which is crucial for accurately representing the physical properties of ice particles in AMPS model.

To address the reviewers' concerns, we have revised the manuscript to include the requested information. Additionally, the reviewer can find the revisions on lines 158-167.

revised lines: 158-167

Within the framework of the AMPS model, ice particles are characterized by their circumscribing sphere volume, denoted as Vcs. This volume is pivotal for comprehending the microphysical behavior of ice particles within mixed-phase clouds. As Hashino and Tripoli (2011a) explains, the circumscribing sphere volume of an ice particle is instrumental in forecasting the mass-dimension (m-D) relationship. The

AMPS model utilizes functions that interlink the predicted sphere volume with the diagnosed aspect ratios, semi-axis lengths, and the particle's maximal extent. Each mass bin's circumscribing sphere volume, a key PPV, is integral to these prognostications. The model assumes a consistent geometric form across microphysical processes, transferring the concentration-weighted circumscribing sphere volume between mass bins as per the collection process. This transfer is essential to ascertain the circumscribing sphere volume of representative hydrometeors, thereby ensuring an accurate representation of the ice particles' physical properties. Moreover, this factor critically influences the determination of the effective parameter.

**RC1-Comment 3:**

Page 5, line 152 Explanation of the reflectivity factor is not accurate. The reflectivity factor characterises a volume of scatterers, not just one object. One object's ability to scatter is characterised by a scattering cross section. To be even more precise Ze is the equivalent reflectivity factor.

Thank you for your insightful observation regarding our explanation of the reflectivity factor on Page 5, Line 152 of our manuscript. We appreciate your clarification and agree that a more precise description is warranted in this context.

In our manuscript, we attempted to describe the reflectivity factor, but we recognize from your comment that our explanation may have inadvertently oversimplified its definition. Indeed, the reflectivity factor  $Z_e$  characterizes a volume of scatterers rather than a single object, and this distinction is crucial in atmospheric sciences, particularly in radar meteorology.

The scattering cross section, as you rightly pointed out, is a more appropriate measure for the scattering ability of an individual object. On the other hand, the equivalent reflectivity factor  $Z_e$  is a more comprehensive parameter that represents the collective scattering properties of a volume containing numerous particles or scatterers. This distinction is fundamental to accurately interpreting radar data, especially in the study of cloud microphysics.

We revised the relevant section in such a way to accurately reflect the abovementioned distinction. The revised text shall now clearly differentiate between the scattering cross section of an individual particle and the equivalent reflectivity factor Ze that characterizes a volume of scatterers. Specifically, the newly added lines 177-181 are as follows:

**New line:**

The equivalent radar reflectivity factor, denoted as  $Z_e$ , more accurately characterizes the collective scattering properties of a volume of scatterers, such as atmospheric precipitation particles, rather than just a single object. This factor is crucial in identifying and quantifying precipitation events by enabling the detection of radar signal reflections from these particles. The magnitude of  $Z_e$  is influenced by several factors: the size and concentration of the precipitation particles, their composition, and the frequency of the radar signal.

**RC1-Comment 4:**

Line 177. *It is a standard practice to use wavelength appropriate value*, **not use one computed for cm-wavelengths**.

Apologies for the misunderstanding. Indeed, PAMTRA uses the value of |Kw| that corresponds to the considered wavelength of a Ka-band radar (8.5 mm, 35.5 GHz). We moved the phrase " (|Kw| = 0.93 at Ka-band; Ulaby et al., 1981)" to line 204, where we introduce Kw.

**RC1-Comment 5:**

Line 273. "...concentration of INP concentration increases."

Thank you for bringing to our attention the redundancy in the phrasing on Line 273 of our manuscript. We appreciate your meticulous review, which helps enhance the precision and clarity of our writing. In accordance with your feedback, we have revised the sentence here:

Revised lines 305-306: In response to the increased concentration of INP, the ice water content increased, as well.

**RC1-Comment 6:**

Line 277. "This reduction in droplet size increases the altitude of the cloud base..." I don't understand why the reduction in droplet size would affect the cloud base. How is the cloud base defined? Is it defined from the LWC profiles? If yes, how do you separate contributions from cloud droplets and drizzle?

Thank you for your insightful comment regarding our description of the cloud base's altitude in relation to droplet size, as mentioned on Line 277 of our manuscript. We realize that our original statement may have lacked the necessary clarity, and we appreciate the opportunity to provide a more detailed explanation.

In our study, the term "cloud base," particularly in the context of mixed-phase clouds, was intended to refer to the lower boundary of the cloud where the concentration of cloud droplets becomes significant. This is typically defined based on the profiles of mixing ratio of water and ice. The mixed-phase cloud base, therefore, encompasses both water and ice crystals. So, we change the line as below, additionally, the reviewer can find the revisions on line 310.

**New line: This reduction in droplet size increases the altitude of the mixed-phase cloud base**

When we mention that "the reduction in droplet size increases the altitude of the cloud base," we aim to describe how microphysical processes, such as the reduction in droplet size due to factors like increased CCN concentration, can lead to changes in the vertical extent of the cloud. This is because smaller droplets are less likely to collide and coalesce into larger droplets, which would otherwise contribute to cloud precipitation and lower the cloud base.

Regarding the distinction between cloud droplets and drizzle in our model, the AMPS utilizes a spectral bin that predicts the size distribution of ice and liquid particles. This approach allows us to account for the entire spectrum of liquid particles, including cloud droplets, drizzle, and rain particles, based on their sizes. While the analysis in our study did not explicitly separate the contributions of cloud droplets from drizzle, it considers all liquid particles in the model.

RC1-Comment 7:

Figure 7. The maximum size of ice particles is just 2 mm for EXP1, while Figure 6 shows significant aggregation. Is there a reason why no larger particles are produced?

Thank you for your query regarding the maximum size of ice particles in EXP1 as presented in Figure 7. Your observation about the discrepancy between the observed maximum particle size and the significant aggregation shown in Figure 6 is indeed an important aspect to address.

The key to understanding this lies in the temporal dynamics of the cloud system in EXP1, as depicted in Figure 5. It is crucial to note that Figure 7 represents average data across a substantial time range, from 100 to 360 minutes. During this period, EXP1's cloud system undergoes notable changes. Specifically, the system becomes unstable, leading to a significant reduction in the number and size of ice particles.

This instability in the cloud system, influenced by the experiment's conditions, iincluding the maximum vertical wind amplitude of +-1m/s, impedes to sustain larger-sized particles. In such a dynamic environment, larger particles that have formed initially due to aggregation processes are less likely to persist or continue growing as the cloud system evolves.

In summary, the maximum particle size observed in EXP1 is a result of the combined effect of the initial aggregation processes and the subsequent dynamic changes in the cloud system, particularly post the 100-minute mark. These factors collectively contribute to the observed limitation in the maximum size of ice particles in this experiment.

**RC1-Comment 8:**

Figure 8. In order to interpret the figure, it would be good to know what AR values for typical particle types in your model are. What are AR values for crystals, graupel and aggregates? It is strange to see that graupel in EXP3 does not produce a noticeable change in AR. The statement "...exhibit a plate-like shape with AR ( $\alpha$ ) < 1," on line 340 is too general and covers a large fraction of different ice particle types, ranging from pristine dendrites to aggregates. It is expected that aggregates have AR around 0.5 – 0.6 range (Hogan et al. 2012; Li et al. 2018 and Matrosov et al. 2017).

Hogan, R. J., L. Tian, P. R. A. Brown, C. D. Westbrook, A. J. Heymsfield, and J. D. Eastment, 2012: Radar Scattering from Ice Aggregates Using the Horizontally Aligned Oblate Spheroid Approximation. *J. Appl. Meteor. Climatol.*, 51, 655–671, https://doi.org/10.1175/JAMC-D-11-074.1.

Li, H., Moisseev, D., & von Lerber, A. (2018). How does riming affect dual-polarization radar observations and snowflake shape? *Journal of Geophysical Research: Atmospheres*, 123, 6070–6081. https://doi.org/10.1029/2017JD028186

Matrosov, S. Y., C. G. Schmitt, M. Maahn, and G. de Boer, 2017: Atmospheric Ice Particle Shape Estimates from Polarimetric Radar Measurements and In Situ Observations. *J. Atmos. Oceanic Technol.*, 34, 2569–2587, https://doi.org/10.1175/JTECH-D-17-0111.1.

Thank you for your insightful comment regarding Figure 8 in our manuscript, specifically your inquiry about the Aspect Ratio (AR) values for different ice particle types such as crystals, graupel, and aggregates in our model. We appreciate the references you provided, which offer valuable context for understanding the expected AR values in various ice particle types. In our paper, we did not delve into the model's details extensively since Hashino, the principal developer of the AMPS model, has elaborated on it in several papers. However, your comments (8, 9, 10, and 11) suggest a need for further elucidation of the model.

To provide a clearer understanding, the AMPS model predicts 16 Particle Property Variables (PPVs) at each time step, as described in response 1. These PPVs allow us to diagnose the type and habit of solid hydrometeors. In our study, we define the "type" of solid hydrometeors based on the physical processes that form them, such as pristine crystal, aggregates, rimed aggregate, graupel, and rimed crystal. This diagnosis is based on mass content components, as illustrated in Fig. 2 by Hashino and Tripoli (2007). For example, graupel is diagnosed if a bin's mass content is predominantly from the riming process. Aggregates are identified when aggregation is the dominant process in a bin's mass content. Similarly, bins containing mass content primarily from vapor deposition are diagnosed as certain pristine crystals. The "habit" of ice crystals refers to their specific crystallographic features, such as plate, dendrite, column, and various polycrystals as shown in Fig 2. The habit diagnosis utilizes length-variable components, aligning with traditional forms found in the Magono–Lee classification. It's important to note that while categorizing solid hydrometeors is not necessary for SHIPS microphysics simulations, it aids in verification as observations are often classified into traditional forms.

Regarding Figure 8, we calculated AR values using the length information of ice particles without segregating by type. This approach was chosen to avoid potential errors that might arise from additional categorization. Of course, in order to compare with other models, it would be better to classify and compare them by the type corresponding to that model. However, since this paper tried to compare from the viewpoint of radar observation, therefore, AR values were computed without type division.

We revised the relevant section 2.2 in such a way to accurately reflect the above mentioned distinction. The revised text shall now clearly differentiate between the ice particle type and the habit. Specifically, the newly added lines 125-137 are as follows:

The identification of ice particle type and habit relies on various components, including mass content, length, and concentration. In our methodology, the SHIPS defines the ice particle model as a conceptual shape to represent ice particles leading to their genesis, encompassing "pristine crystals, aggregates, rimed aggregates, graupel, and rimed crystals" as explicated in Figure 2 by Hashino and Tripoli (2007). Pristine crystals, rimed aggregates, aggregates, and rimed aggregates are modeled as cylinders, while graupel is represented as a spheroid. These shapes serve as the basis for determining the maximum dimension (D) of each particle. However, it is important to note that this diagnosis is primarily intended for comparison with observations or other models using predicted mass bin information within the PPVs. Therefore, additional errors may arise when artificially categorizing these types. In this study, we analyzed the mass bin information without separate type divisions for a more comprehensive assessment.

The habit of ice crystals, a critical aspect of our study, is determined by analyzing their unique crystallographic properties which include forms such as plates, dendrites, columns, and three polycrystals, as illustrated in Figure 1.

**RC1-Comment 9:**

Line 351 "Figure 9(a) provides insights into the predominant shape of ice particles, with plates representing the majority of the particles." What do you mean when you state that plates represent the majority of the particles? Are aggregates plates? What about graupel? I think you need to use more precise terminology.

Thank you for your insightful comment on Line 351 of our manuscript, specifically regarding our discussion of the predominant ice particle shapes in Figure 9(a). We recognize the importance of precise terminology in conveying the complex microphysical characteristics of ice particles, especially in a nuanced field like cloud microphysics.

In our initial statement that "plates represent the majority of the particles," our objective was to highlight the predominance of plate-shaped ice particles under the specific atmospheric conditions we simulated. Your question rightly calls for a clearer distinction between different ice particle types and process-based information. As seen in response 1, all the PPV variables are listed.

To clarify our approach, we utilized the Spectral Ice Habit Prediction System (SHIPS) within the AMPS model, which allowed us to categorize ice particles into distinct shapes like plates, dendrites, columns, and three types of polycrystals, as explained in Figures 1 and 2. This detailed categorization, based on particle shape information, is explained in Section 2.2. The reference to "plates represent the majority of the particles" was intended to convey that in the modeled scenario, plate-shaped particles were the most prevalent form identified by SHIPS, reflecting the specific microphysical conditions at play. This does not negate the presence or importance of other microphysical processes, such as those involving aggregates, which are also integral components of the AMPS model.

We appreciate your point about the potential confusion arising from differentiating process-based information from particle shape information. To enhance clarity in our manuscript, we included a more detailed explanation linking microphysical processes (such as riming, aggregation, and crystal growth) to the specific particle types identified by SHIPS in AMPS. This shall help elucidate how these processes manifest in various particle shapes within mixed-phase clouds. The AMPS model's detailed output, while complex, provides a robust platform for a comprehensive analysis of cloud microphysics. In our study, we leveraged this detail to analyze and present a multifaceted view of cloud particle characteristics and processes.

We revised the relevant section 2.2 in such a way to accurately reflect the above mentioned distinction. The revised text shall now clearly differentiate between the ice particle type and the habit. Specifically, the newly added lines 125-146 vare as follows:

**new lines: 125-146**

The identification of ice particle type and habit relies on various components, including mass content, length, and concentration. In our methodology, the SHIPS defines the ice particle model as a conceptual shape to represent ice particles leading to their genesis, encompassing "pristine crystals, aggregates, rimed aggregates, graupel, and rimed crystals" as explicated in Figure 2 by Hashino and Tripoli (2007). Pristine crystals, rimed aggregates, aggregates, and rimed aggregates are modeled as cylinders, while graupel is represented as a spheroid. These shapes serve as the basis for determining the maximum dimension (D) of each particle. However, it is important to note that this diagnosis is primarily intended for comparison with observations or other models using predicted mass bin information within the PPVs. Therefore, additional errors may arise when artificially categorizing these types. In this study, we analyzed the mass bin information without separate type divisions for a more comprehensive assessment.

The habit of ice crystals, a critical aspect of our study, is determined by analyzing their unique crystallographic properties which include forms such as plates, dendrites, columns, and three polycrystals, as illustrated in Figure 1. For each identified particle habit, the SHIPS within the AMPS model assigns an ice particle model that represents the geometric shape enveloping the ice particle. This model encompasses detailed crystal habit information-such as the a-axis length (la), representing the radius; the c-axis length (lc), representing the height; and the d-axis length (ld), representing the dendritic arm-across the three crystal habits of plate, columnar, and dendrite for monocrystals. Additionally, the model employs a PPV, termed the extra crystalline structure number (nexice), which ranges from 0 to 1. A value of nexice greater than or equal to 0.5 signifies that the ice crystals in a particular bin are polycrystals. The SHIPS's methodical approach also integrates the coordinates of the center of gravity (ag, cg), measured along the a- and c-axes from the center of the monocrystals, as distinct PPVs. These measurements are pivotal in differentiating between planar and columnar polycrystals: a planar polycrystal is identified if the ratio ag/la exceeds the ratio cg/lc by more than 0.5, whereas a columnar polycrystal is determined if cg/lc exceeds ag/la by more than 0.5. Ice crystals that do not fit within these criteria, such as scale-like side planes, are categorized as irregular polycrystals. The process and criteria for habit diagnosis are further detailed in Figure 2.

**RC1-Comment 10:**

Also, in the figure you use "pploy, cploy and irploy", instead of "ppoly, cpoly and irpoly".

Thank you for pointing out the typographical errors in the labels of Figure 9. We corrected them.

**RC1-Comment 11:**

Line 364. While it is true that in Figure 9 we can see the difference between AR values for different experiments, I find it difficult to make a connection between this figure and Fig. 6. In Fig. 6 you use rime, aggr. and crystal. But later you introduce a very different classification: namely plate, dendrite, column, and various types of polycrystals in Fig. 9 (a) and in Fig 9 (b) this classification is reduced to plate and ir. poly. It would be better if you would be consistent. In Fig. 10 you go back to riming aggregation and deposition growth, so to which particle in Fig. 9 these processes correspond to?

Thank you for your insightful comment on Line 364 concerning the classifications used in Figures 6, 9, and 10 of our manuscript. We appreciate your attention to detail and the opportunity to clarify our approach.

In Figure 6, we utilized terms such as 'rim', 'agg.', and 'crystal' primarily for simplification, aiming to provide a broad, accessible understanding of the key microphysical processes influencing ice particle formation and growth within clouds.

Moving to Figure 9, we introduced a more detailed analysis using particle shape classification - 'plate', 'dendrite', 'column', and various 'polycrystals' - reflecting the SHIPS within the AMPS model. We acknowledge that particle shape information could potentially cause confusion in correlating microphysical processes. To address this and clarify the connections, we enhanced the manuscript by including a more detailed explanation of the relationship between the process based mass information and the specific particle types as characterized by SHIPS in AMPS. See additional lines 125-137 in the section 2.2 of the revised manuscript.

The AMPS model's comprehensive detail, while occasionally complex, is indeed its strength, offering an in-depth analysis of cloud microphysics. We endeavored to harness this advantage by analyzing the model's outputs from various perspectives.

RC1-Comment 12:

Figure 12. Looking at EXP1 reflectivity profile in this figure, I find it even more surprising that in Fig. 7 the maximum particle size for EXP1 was just 2 mm. I would have expected to see larger aggregates.

Thanks for this comments. As pointed out in Response 7, we think that the environmental conditions prescribed in the model setup did not allow for a growth of particles beyond 2 mm. We however cross-checked the reflectivity calculation and confirmed that the results shown in Fig. 12 do well correspond to the contrasts in the size distribution.

This study presents modeling the impact of CCN and INP on the mixed-phased clouds, and the application of the model to the radar simulator. Since the CCN and INP are important contributors to cloud formation, the study will help to improve our understanding of formation of mix-phased clouds and the controlling factors. The manuscript is well-written and should be publishable after the following issues are resolved:

**RC2-Comment 1:**

The title doesn't seem to reflect the content of this study which mainly focuses on modeling the impact of CCN and INP concentrations on the mixed-phase clouds. It is suggested to change the title for more precision on the subject.

Thank you for your valuable feedback regarding the title of our manuscript. We appreciate your keen observation and agree that the title should accurately and comprehensively reflect the main focus and findings of our study. Your suggestion highlights the importance of precision and clarity in effectively communicating the core subjects to the readers.

In our study, while modeling the impacts of CCN and INP on mixed-phase clouds is indeed a primary focus, we also dedicate a significant portion of our work to analyzing results using a radar forward simulator. This approach in one of our chapters allows us to directly compare our modeling outcomes with observational data, providing a critical validation and offering insights into the real-world applicability and accuracy of our models.

Considering the multifaceted nature of our study, which combines detailed modeling with observational comparisons, we propose the following revised titles to more accurately represent the content and emphasis of our research:

**candidate title: "Simulations of the impact of CCN and INP perturbations on the microphysics and radar reflectivity factor of stratiform mixed-phase clouds**

We believe that the proposed changes provide a more precise and encompassing representation of our study's content and focus. We have amended the manuscript to include the revised title that best encapsulates our research.

**RC2-Comment 2:**

It would be beneficial to the broader readers if the authors can provide some explanation on why the initial profiles of the potential temperature and specific humidity look like the ones shown in Figure 3.

Thank you for your insightful comment and the opportunity to clarify the aspects related to the initial profiles of potential temperature and specific humidity as depicted in Figure 3 of our manuscript.

Importantly, the profiles are designed as a test of a mixed-phase stratocumulus layer, with the profile taken from the GCSS (Global Energy and Water Cycle Experiment Cloud System Study) SHEBA (Surface Heat Budget of the Arctic Ocean) intercomparison as shown in previous studies (e.g., Klein et al, 2009; Morrison et al., 2011, Fridlind et al., 2012). This choice was made to ensure that our study's conditions align with well-established benchmarks and facilitate comparisons with other studies focusing on similar Arctic environments. The GCSS SHEBA intercomparison provides a well-documented set of atmospheric conditions that are widely recognized as representative of Arctic stratocumulus clouds, making it an ideal source for our initial profiles.

The initial profiles for both potential temperature and specific humidity are fundamental in setting the stage for the subsequent cloud microphysical and dynamical processes explored in our study. They are derived from a combination of observational data, theoretical understanding, and previous modeling studies focused on Arctic mixed-phase clouds. We believe that these initial conditions are representative of the typical Arctic environment and are crucial for the accuracy and reliability of our simulation results.

We added lines 227-235 to the revised manuscript in order to include the main points of the above-written explanation into our publication.

**New lines:**

Importantly, the profiles were specifically crafted to emulate a mixed-phase stratocumulus layer, drawing inspiration from the GCSS (Global Energy and Water Cycle Experiment Cloud System Study) SHEBA (Surface Heat Budget of the Arctic Ocean) intercomparison, a choice substantiated by previous studies (Klein et al., 2009; Morrison et al., 2011; Fridlind et al., 2012). This selection was made with the deliberate intent of aligning our study's environmental conditions with well-established benchmarks, thereby enabling seamless comparisons with other investigations centered around similar Arctic settings. The GCSS SHEBA intercomparison is widely acknowledged for providing meticulously documented atmospheric conditions that faithfully represent Arctic stratocumulus clouds, rendering it an ideal resource for our initial profiles.

**RC2-Comment 3:**

The effective diameter for EXP2 shown in Figure 4 is not 193 (page 10), judged from the figure.

Thank you for pointing out the discrepancy regarding the effective diameter for EXP2 as mentioned on page 10 and depicted in Figure 4. Upon re-examining the figure and our data, we have identified a typographical error in our manuscript. The correct effective diameter for EXP2 is indeed 293, not 193 as initially stated. We deeply appreciate your sharp observation and thank you for bringing this to our attention. We corrected it the line 287 of the revised manuscript.

**RC2-Comment 4:**

For EXP1, high INPs lead to not only a spread of smaller particles but also large particles (over 1 mm) as shown in Figure 7. Can the authors give a likely reason for this spread of larger particles?

Thank you for pointing out the spread of larger particles in EXP1, as depicted in Figure 7. This observation is crucial for understanding the microphysics of clouds under high Ice Nucleating Particle (INP) conditions. The phenomenon observed in EXP1, characterized by the presence of larger particles, can be attributed to several interrelated processes.

Firstly, the high concentration of INPs in EXP1 leads to enhanced nucleation, significantly increasing the formation of ice nuclei and, consequently, the number of initial ice crystals. This initial surge in nucleation is evident in Figure 1, where we see a substantial increase in ice crystal numbers, followed by a gradual decrease. This increased number of ice particles sets the stage for their growth through subsequent processes.

As more ice particles form, the likelihood of their collision and aggregation rises. This interaction, especially under the conditions of EXP1, leads to the formation of larger ice particles. The aggregation contributes to a gradual increase in the size of these ice particles, resulting in the observed spread of larger particles.

Additionally, Figure 5 indicates that EXP1's cloud system becomes unstable over time, leading to a marked reduction in ice particles that subsequently fall. Following this period of instability, there is a resurgence of smaller ice particles in the upper layer after 200 minutes. When we combine and average the data after 100 minutes, we observe the coexistence of both large ice particles and newly formed smaller ones in EXP1. This pattern suggests a dynamic cloud environment where particle sizes are influenced by evolving microphysical processes.

In summary, the unique conditions in EXP1, characterized by high INP concentrations, lead to a complex interplay of nucleation, aggregation, and collision processes, alongside dynamic changes in cloud stability and limited LWC. These factors collectively result in the spread of both larger and smaller ice particles, as observed in our experimental data.

We added lines 501-505 to the revised manuscript in order to include the main points of the above-written explanation into our publication.

**New lines 501-505**

However, a significant presence of irregular polycrystals is observed, especially in scenarios with high INP concentrations. In scenarios with high INP concentrations, we observe a complexinterplay of nucleation, aggregation, and collision phenomena. These occur against a backdrop of dynamic changes in cloud stability and limited LWC. Collectively, these processes lead to a broader distribution of ice particle sizes, ranging from larger to smaller, a pattern that is corroborated by our experimental data. The AR of ice particles is affected by both CCN and INP concentrations, with higher concentrations leading to smaller AR values.

**RC2-Comment 5:**

When accounting for the impact of INPs on the AR, the authors point out that the fluctuations of AR were due to formation of irregular polycrystals after 200 minutes. Two things need to be clarified: 1) why does this occur after 200 minutes? 2) what would happen if the simulation time was extended longer, for example, AR will be more stable?

Thank you for your perceptive inquiry regarding the impact of Ice Nucleating Particles (INPs) on the Aspect Ratio (AR) and the formation of irregular polycrystals in our simulation. We appreciate the opportunity to provide further clarification on these aspects:

Regarding the formation of irregular polycrystals around the 200-minute mark, this occurrence is primarily driven by the interplay between the initial conditions, dynamics, and microphysics of the simulation. The appearance of irregular polycrystals at approximately 200 minutes in our experiment is a result of these specific conditions. However, it's important to note that this timing is not fixed and could vary depending on the initial conditions and dynamic factors in different simulations. The 200-minute mark is indicative in our specific experiment but is not a universal threshold for the formation of these structures.

As for extending the simulation time beyond the current duration, it is unlikely that the AR would stabilize in EXP1. This is because the dynamics of the cloud system in this particular experiment are not directly influenced by microphysical changes. As detailed in our initial experimental setup, a continuous vertical wind and consistent humidity conditions were maintained throughout. Therefore, if the simulation time were extended, we would likely see a repetition of the pattern where an increased AR eventually decreases again. This cyclical pattern is a characteristic of the cloud system dynamics under the specific conditions of EXP1.

We acknowledge that these complex interactions between dynamics and microphysics are crucial for understanding cloud behavior and we added in the summary section lines 531-536 to the revised manuscript in order to include the main points of the above-written explanation into our publication.

New lines 531-536:

Moreover, the occurrence of irregular polycrystals at approximately the 200 minute mark highlights the intricate interplay between the initial conditions, dynamics, and microphysics of the simulation in the High INP case. This particular timing, which is unique to our experiment, demonstrates the susceptibility of ice particle formation processes to different conditions. It is noteworthy that, given the consistent vertical wind and humidity conditions established in our simulation, extending the duration would likely result in the persistence of observed cyclical AR patterns, rather than a stabilization of AR values.

**RC2-Comment 6:**

As shown in Figure 12, the trend of the impact of INPs on the reflectivity is surprisingly reversed from decrease to increase. The explanation for this transition seems not satisfactory. I wonder if more refining experiments were performed, for example, smaller steps of INPs were set, this transition might be observable.

Thank you for highlighting the trend of the impact of Ice Nucleating Particles (INPs) on reflectivity as depicted in Figure 12. Your observation about the surprising reversal in this trend, transitioning from a decrease to an increase in reflectivity, is a crucial point that indeed merits a more detailed explanation.

Upon reviewing Figure 12 alongside Table 2, it becomes apparent that as INP concentrations increase, there is a corresponding rise in the average reflectivity values from -11.83 to -10.09, and then a notable jump to 4.65. This pattern suggests a continuous increase rather than a decrease followed by an increase. The trend observed in Figure 12 is consistent and continuous, indicating a direct correlation between increasing INP concentrations and higher reflectivity values.

This phenomenon can be attributed to the complex interactions between INPs and cloud microphysical processes. Higher concentrations of INPs tend to facilitate more ice nucleation, leading to an increase in ice particle numbers and sizes. These larger and more numerous ice particles contribute to higher reflectivity values.

We recognize the potential value of conducting more refined experiments with smaller increments of INP concentrations to further explore this transition. Experiments should be performed in the future that deal with a more granular distribution of INP and CCN perturbations. Nevertheless, as pointed out in the introduction, the CCN and INP concentrations applied in our study are motivated by realistic values which can occur under ambient conditions in contrasting regions of the globe. Thus, an important question remains from our study. Given that AMPS is well capable of dealing with the applied INP and CCN ranges, there might indeed be strong effects of aerosol perturbations present on mixed-phase clouds. On the other hand, if the model cannot handle such strong yet realistic contrasts, fundamental theories or parameterizations of mixed-phase cloud processes in state-of-the-art models like AMPS need to be re-evaluated and refined. Further investigations should be performed on how aerosol perturbations affect the evolution of idealized stratiform mixed-phase clouds. It is necessary to understand and evaluate these basic processes before an accurate general assessment of aerosol-cloud interaction can be realized.

**Table 1.** The table shows the 16 advected prognostic variables representing the particle property variables (PPVs) within a liquid and ice spectra across each bin, as utilized in the KiD-AMPS model. Here,  $\rho_m$  denotes the moist air density, while  $\rho_{lat}$  and  $\rho_{iat}$  specify the total aerosol density within the liquid phase and ice phase, respectively. Additionally,  $\rho_{las}$  and  $\rho_{ias}$  correspond to the soluble aerosol density in the liquid and ice phases.

[revised manuscript text omitted]